# Towards Generalizable Multi-Camera 3D Object Detection via Perspective Debiasing

## Abstract

Detecting objects in 3D space using multiple cameras, known as Multi-Camera 3D Object Detection (MC3D-Det), has gained prominence with the advent of bird's-eye view (BEV) approaches. However, these methods often struggle when faced with unfamiliar testing environments due to the lack of diverse training data encompassing various viewpoints and environments. To address this, we propose a novel method that aligns 3D detection with 2D camera plane results, ensuring consistent and accurate detections. Our framework, anchored in perspective debiasing, helps the learning of features resilient to domain shifts. In our approach, we render diverse view maps from BEV features and rectify the perspective bias of these maps, leveraging implicit foreground volumes to bridge the camera and BEV planes. This two-step process promotes the learning of perspective- and context-independent features, crucial for accurate object detection across varying viewpoints, camera parameters and environment conditions. Notably, our model-agnostic approach preserves the original network structure without incurring additional inference costs, facilitating seamless integration across various models and simplifying deployment. Furthermore, we also show our approach achieves satisfactory results in real data when trained only with virtual datasets, eliminating the need for real scene annotations. Experimental results on both Domain Generalization (DG) and Unsupervised Domain Adaptation (UDA) clearly demonstrate its effectiveness. Our code will be released.

## 1 Introduction

Multi-Camera 3D Object Detection (MC3D-Det) refers to the task of detecting and localizing objects in 3D space using multiple cameras (Ma et al., 2022; Li et al., 2022a). By combining information from different viewpoints, multi-camera 3D object detection can provide more accurate and robust object detection results, especially in scenarios where objects may be occluded or partially visible from certain viewpoints. In recent years, bird 's-eye view (BEV) approaches have gained tremendous attention for the MC3D-Det task (Ma et al., 2022; Li et al., 2022a; Liu et al., 2022; Wang et al., 2022). Despite their strengths in multi-camera information fusion, these methods may face severe performance degeneration when the testing environment is significantly different from the training ones.

Two promising directions to alleviate the distribution shifts are domain generalization (DG) and unsupervised domain adaptation (UDA). DG methods often decouple and eliminate the domain-specific features, so as to improve the generalization performance of the unseen domain (Wang et al., 2023a). Regarding to UDA, recent methods alleviate the domain shifts via generating pseudo labels eccv2022uda, $Yuan_2023_C VPR$, $Yang_2021_C VPR$ or latent feature distribution alignment $Xu_2023_C VPR$, $yan2023ss$ and environment $-$ independent features.

Our observations indicate that 2D detection in a single-view (camera plane) often has a stronger ability to generalize than multi-camera 3D object detection, as shown in Fig. 1. Several studies have explored the integration of 2D detection into MC3D-Det, such as the fusion of 2D information into 3D detectors (Wang et al., 2023d; Yang et al., 2023) or the establishment of 2D-3D consistency (Yang et al., 2022; Lian et al., 2022). Fusing 2D information is a learning-based approach rather than a mechanism modeling approach and is still significantly affected by domain migration. Existing 2D-3D consistency methods project 3D results onto the 2D plane and establish consistency.

Such constraints can compromise the semantic information in the target domain rather than modifying the geometric information. Furthermore, this 2D-3D consistency approach makes having a uniform approach for all detection heads challenging. For instance, in the case of centerpoint (Yin et al., 2021), the final prediction is obtained by combining the outputs of the classification head and an offset regression head.

To this end, we introduce a perspective debiasing framework to modify the feature geometry position of BEV space directly, enabling the learning of perspective- and context-invariant features against domain shifts. Our approach involves two main steps: 1) rendering diverse view maps from BEV features and 2) rectifying the perspective bias of these maps. The first step leverages implicit foreground volumes (IFV) to relate the camera and BEV planes, allowing for rendering view maps with varied camera parameters. The second step, in the source domain, uses random camera positions and angles to supervise the camera plane map rendered from IFV, promoting the learning of perspective- and context-independent features. Similarly, in the target domain, a pre-trained 2D detector aids in rectifying BEV features. Notably, our model-agnostic approach preserves the original network structure without incurring additional inference costs, facilitating seamless integration across various models and simplifying deployment. This reduces development and maintenance complexity and ensures efficiency and resource conservation, which are crucial for real-time applications and long-term, large-scale deployments.

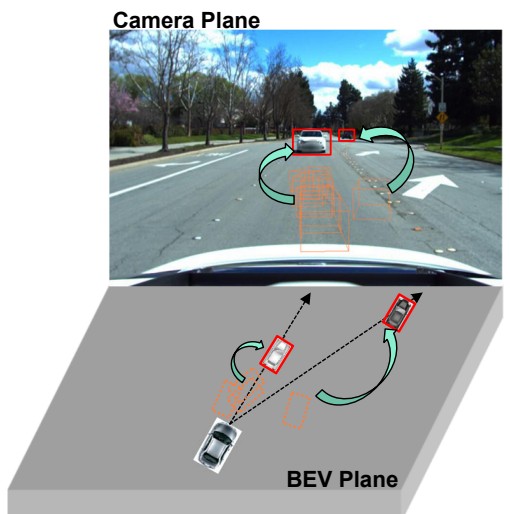

Figure 1: Domain gap challenges cause MC3D-Det sometimes to produce spurious and deteriorate depth estimations. By contrast, 2D detectors typically demonstrate more precise performance against domain gap, suggesting potential strategies to adjust 3D detector inaccuracies.

We established the UDA benchmark on MC3D-Det to verify our method and instantiated our framework on BEVDepth, achieving excellent results in both DG and UDA protocol. We also pioneer the use of training on virtual datasets, bypassing the need for real scene annotations, to enhance real-world multi-camera 3D perception tasks. In summary, the core contributions of this paper are:

- We propose a generalizable MC3D-Det framework based on perspective debiasing, which can not only help the model learn the perspective- and context-invariant feature in the source domain but also utilize the 2D detector further to correct the spurious geometric features in the target domain.

- We make the first attempt to study unsupervised domain adaptation on MC3D-Det and establish a benchmark. Our approach achieved the state-of-the-art results on both UDA and DG protocols.

- We explore the training on virtual engine without the real scene annotations to achieve real-world MC3D-Det tasks for the first time.

## 2 RELATED WORKS

### 2.1 VISION-BASED 3D OBJECT DETECTION

Multi-camera 3D object detection (MC3D-Det) targets to identify and localize objects in 3D space, received widespread attention (Ma et al., 2022; Li et al., 2022a). Recently, most of MC3D-Det methods extract image features and project them onto the bird 's-eye view (BEV) plane for better integrating the spatial-temporal feature. Orthographic feature transform (OFT) and Lift-splat-shoot (LSS) provide the early exploration of mapping the multi-view features to BEV space (Roddick et al., 2019; Philion & Fidler, 2020). Based on LSS, BEVDet enables this paradigm to the detection task competitively (Huang et al., 2021; Li et al., 2023b;a). BEVformer further designs a transformer

structure to automatically extract and fuse BEV features, leading to excellent performance on 3D detection (Li et al., 2022c). PETR series propose 3D position-aware encoding to enable the network to learn geometry information implicitly (Liu et al., 2022; 2023). These methods have achieved satisfactory results on the in-distribution dataset but may show very poor results under cross-domain protocols.

## 2.2 CROSS DOMAIN PROTOCOLS ON DETECTION

Domain generalization or unsupervised domain adaptation aims to improve model performance on the target domain without labels. Many approaches have been designed for 2D detection, such as feature distribution alignment or pseudo-label methods (Muandet et al., 2013; Li et al., 2018; Dou et al., 2019; Facil et al., 2019; Chen et al., 2018; Xu et al., 2020; He & Zhang, 2020; Zhao et al., 2020). These methods can only solve the domain shift problem caused by environmental changes like rain or low light. For the MC3D-Det task, there is only one study for domain shift, which demonstrates that an important factor for MC3D-Det is the overfitting of camera parameters (Wang et al., 2023a). Essentially, the fixed observation perspective and similar road structures in the source domain lead to spurious and deteriorated geometric features. However, without additional supervision, it is very difficult to further extract perspective- and context-independent features on the target domain.

## 2.3 VIRTUAL ENGINE FOR AUTOMATIC DRIVING

Virtual engines can generate a large amount of labeled data, and DG or UDA can utilize these virtual data to achieve the perception of real scenes. According to the requirements, the virtual engine has better controllability and can generate various scenarios and samples: domain shift (Sun et al., 2022), vehicle-to-everything (Xu et al., 2022; Li et al., 2022b), corner case (Kim et al., 2022; Wang et al., 2023c). So, breaking the domain gap between virtual and real datasets can further facilitate the closed-loop form of visually-oriented planning (Jia et al., 2023). To our best knowledge, there are no studies that only use virtual engine without real scenes labels for MC3D-Det.

## 3 PRELIMINARIES

### 3.1 PROBLEM SETUP

Our research is centered around enhancing the generalization of MC3D-Det. To achieve this goal, we explore two widely used and practical protocols, namely, domain generalization (DG) and unsupervised domain adaptation (UDA).

● For DG on MC3D-Det task, our primary objective is to leverage solely the labeled data from the source domain $D_S = \{X_s^i, Y_s^i, K_s^i, E_s^i\}$ to improve the generalization of model. Here, the $i$-th sample contains $N$ multi view images $X^i = \{I_1, I_2, ..., I_N\}$ (superscript is omitted for clearity) and the corresponding intrinsic $K^i$ and extrinsic parameters $E^i$ of camera. The labels of source domain $Y_s^i$ includes location, size in each dimension, and orientation.

● For UDA on MC3D-Det task, additional unlabeled target domain data $D_T = \{X_t^i, K_t^i, E_t^i\}$ can be utilized to further improve the generalization of model. The only difference between DG and UDA is whether the unlabeled data of the target domain can be utilized.

### 3.2 PERSPECTIVE BIAS

To detect the object's location $L = [x, y, z]$ at the BEV space, corresponding to the image plane $[u, v]$, most MC3D-Det methods involves two essential steps: (1) get the the image features from the j-th camera by the image encoder $F_{img}$. (2) map these feature into BEV space and fuse them to get the final location of objects by BEV encoder $F_{bev}$:

$$
\begin{aligned}
L &= F_{bev}(F_{img}(I_1), ..., F_{img}(I_N), K, E) \\
&= L_{gt} + \Delta L_{img} + \Delta L_{bev},
\end{aligned}
\tag{1}
$$

where $L_{gt}$, $\Delta L_{img}$ and $\Delta L_{bev}$ are the ground-truth location and the bias of img encoder ($F_{img}$) and BEV encoder ($F_{bev}$). Both $\Delta L_{img}$ and $\Delta L_{bev}$ are caused by overfitting limited viewpoint,

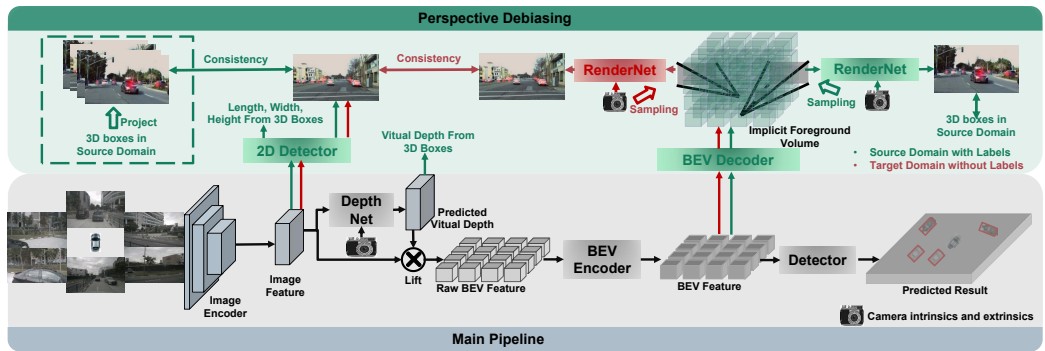

Figure 2: The generalizable framework (PD-BEV) based on perspective debiasing. The main pipeline of BEVDepth is shown in the bottom part of the figure. With the supervision of heatmaps and virtual depth, the semantic and geometric knowledge is injected into preliminary image features in advance. Then, implicit foreground volume (IFV) is tailored as a carrier for the camera plane and the BEV plane. The rendered heatmaps from IFV are supervised by 3D boxes in the source domain and by the pre-trained 2D detector in the target domain. The green flow means the supervision of the source domain and the red flow is for the target domain. The RenderNet shares the same parameters.

camera parameter, and similar environments. Without additional supervision in the target domain, $\Delta L_{img}$ and $\Delta L_{img}$ are difficult to be mitigated. So we turn the space bias into the bias of a single perspective. We show the perspective bias $[\Delta u, \Delta v]$ on the uv image plane as:

$$[\Delta u, \Delta v] = = [\frac{k_u(u - c_u) + b_u}{d(u, v)}, \frac{k_v(v - c_v) + b_v}{d(u, v)}]. \tag{2}$$

where $k_u$, $b_u$, $k_v$, and, $b_v$ is related to the domain bias of BEV encoder $\Delta L_{BEV}$, and $d(u, v)$ represents the final predicted depth information of the model. $c_u$ and $c_v$ represent the coordinates of the camera's optical center in the uv image plane. The detail proof and discussion in appendix C. Eq. 2 provides us with several important inferences: (1) the presence of the final position shift can lead to perspective bias, indicating that optimizing perspective bias can help alleviate domain shift. (2) Even points on the photocentric rays of the camera may experience a shift in their position on the uv image plane.

Intuitively, the domain shift changes the BEV feature position and value, which arises due to over-fitting with limited viewpoint and camera parameters. To mitigate this issue, it is crucial to re-render new view images from BEV features, thereby enabling the network to learn perspective- and environment-independent features. In light of this, the paper aims to address the perspective bias associated with different rendered viewpoints to enhance the generalization ability of the model.

## 4 METHOD

To reduce bias stated in Eq. 2, we tailored a generalizable framework (PD-BEV) based on perspective debiasing as shown in Fig. 2. Our framework is model-agnostic, and we demonstrate its effectiveness by optimizing BEVDepth as an example.

### 4.1 SEMANTIC RENDERING

We first introduce how to establish the connection between 2D image plane and BEV space. However, most MC3D-Det methods utilize the BEV plane representations without height dimension (Huang et al., 2021; Li et al., 2023a;b), so we propose the implicit foreground volume for rendering new viewpoints. Specifically, we use a geometry-aware decoder $D_{geo}$ to transform the BEV feature $F_{bev} \in \mathbb{R}^{C \times X \times Y}$ into the intermediate feature $F'_{bev} \in \mathbb{R}^{C \times 1 \times X \times Y}$ and $F_{height} \in \mathbb{R}^{1 \times Z \times X \times Y}$, and this feature is lifted from BEV plane to an implicit foreground volume $V_{ifv} \in \mathbb{R}^{C \times Z \times X \times Y}$:

$$V_{ifv} = \text{sigmoid}(F_{height}) \cdot F_{bev}. \tag{3}$$

Eq. 3 lifts the object on the BEV plane into 3D volume with the estimated height position $sigmoid(F_{height})$. $sigmoid(F_{height})$ represents whether there is an object at the corresponding height. Where XYZ is the three-dimensional size of the established BEV feature. Ideally, the volumes $V_{ifv}$ contain all the foreground objects information in the corresponding position.

To render semantic features of different viewpoints, we propose the Multi-View Semantic Rendering (MVSR). Specifically, we first randomly perturb the camera's position $(x + \triangle x, y + \triangle y, z + \triangle z)$ and orientation $(\theta_{yaw} + \triangle\theta_{yaw}, \theta_{pitch} + \triangle\theta_{pitch}, \theta_{roll} + \triangle\theta_{roll})$. Based on the camera's position and observation orientation, we generate the coordinate of multiple rays $r_i^{w,h} = [x^{w,h}, y^{w,h}, z^{w,h}]$ to sample from implicit foreground volumes $V_{ifv}$ and aggregate them into the camera plane feature $F_{render}$:

$$F(w,h)_{render} = \sum_{i=1}^{n} V_{ifv}(x^{w,h}, y^{w,h}, z^{w,h}), \qquad (4)$$

where $r_i^{w,h} = [x^{w,h}, y^{w,h}, z^{w,h}]$ represents the ray coordinates of $w$-th row and $h$-th column camera plane in the implicit foreground volumes $V_{ifv}$. The rendered camera plane feature $F_{render}$ is then fed into the RenderNet $R$, which is the combination of several 2D convolutional layers, to generate the heatmaps $h_{render} \in \mathbb{R}^{N_{cls} \times W \times H}$ and attributes $a_{render} \in \mathbb{R}^{N_{cls} \times W \times H}$. $N_{cls}$ means the number of categories. The detailed structure of RenderNet is introduced in appendix B.4. The semantic heatmaps and attributes can be constrained on the source and target domains to eliminate perspective bias $[\Delta u, \Delta v]$.

## 4.2 PERSPECTIVE DEBIASING ON SOURCE DOMAIN

To reduce perspective bias as stated in Eq. 2, the 3D boxes of source domain can be used to monitor the heatmaps and attributes of new rendered view. In addition, we also utilize normalized depth information to help the image encoder learn better geometry information.

### 4.2.1 PERSPECTIVE SEMANTIC SUPERVISION

Based on Sec. 4.1, the heatmaps and attributes from different perspectives (the output of RenderNet) can be rendered. Here we will explain how to regularize them to eliminate perspective bias Eq. 2. Specifically, we project the object's box from ego coordinate to the $j$-th 2D camera plane using the intrinsic $K'_j$ and extrinsic parameters $E'_j$ of the rendering process: $\hat{P}_j = (ud, vd, d) = K'_j E'_j P$, where $\hat{P}_j$ and $P$ stand for the object on 2.5D camera plane and 3D space, $d$ represents the depth between the object and the view's optical center. Based on the position of the object on the image plane, the category heatmaps $h_{gt} \in \mathbb{R}^{N_{cls} \times W \times H}$ can be generated (Yin et al., 2021). The object's dimensions (length, width and height) $a_{gt} \in \mathbb{R}^{N_{cls} \times W \times H}$ are also projected to the uv plane. Following (Yin et al., 2021), focal loss $Focal()$ (Lin et al., 2017) and L1 loss $L1$ are used to supervise the class information and object dimensions on source domain:

$$\mathcal{L}_{render} = Focal(h_{render}, h_{gt}) + L1(a_{render}, a_{gt}). \qquad (5)$$

Additionally, we also train a 2D detector for the image feature using 3D boxes by $\mathcal{L}_{ps}$, which uses the same mapping and supervision methods as above. The only difference is that the 3D boxes are projected using the original intrinsics $K$ and extrinsics $E$ of the camera. 2D detectors can be further applied to correct the spurious geometry in the target domain.

### 4.2.2 PERSPECTIVE GEOMETRY SUPERVISION

Providing the explicit depth information can be effective in improving the performance of multi-camera 3D object detection (Li et al., 2023a). However, the depth of the network prediction tends to overfit the intrinsic parameters. So, following (Park et al., 2021; Wang et al., 2023a), we force the DepthNet to learn normalized virtual depth $D_{virtual}$:

$$\mathcal{L}_{pg} = BCE(D_{pre}, D_{virtual}),$$
$$D_{virtual} = \frac{\sqrt{\frac{1}{f_u^2} + \frac{1}{f_v^2}}}{U} D, \qquad (6)$$

where $BCE()$ means Binary Cross Entropy loss, and $D_{pre}$ represents the predicted depth of Depth-Net. $f_u$ and $f_v$ are of v and v focal length of image plane, and $U$ is a constant. It is worth noting that the depth $D$ here is the foreground depth information provided using 3D boxes rather than the point cloud. By doing so, The DepthNet is more likely to focus on the depth of foreground objects. Finally, when using the actual depth information to lift semantic features into BEV plane, we use Eq. 6 to convert the virtual depth back to the actual depth.

### 4.3 PERSPECTIVE DEBIASING ON TARGET DOMAIN

Unlike the source domain, there are no 3D labels in the target domain, so the $\mathcal{L}_{render}$ can't be applied. Subtly, the pre-trained 2D detector is utilized to modify spurious geometric BEV feature on the target domain. To achieve this, we render the heatmaps $h_{render}$ from the implicit foreground volume with the original camera parameters. Focal loss is used to constrain the consistency between the pseudo label of 2D detector and rendered maps:

$$\mathcal{L}_{con} = Focal(h_{render}, h_{pseudo}),$$
$$h_{pseudo} = \left\{ \begin{array}{ll} 1, & h > \tau \\ h, & else \end{array} \right., \tag{7}$$

where $Focal(,)$ is original focal loss (Lin et al., 2017). $\mathcal{L}_{con}$ can effectively use accurate 2D detection to correct the foreground target position in the BEV space, which is an unsupervised regularization on target domain. To further enhance the correction ability of 2D predictions, we enhanced the confidence of the predicted heatmaps by a pseudo way.

### 4.4 OVERALL FRAMEWORK

Although we have added some networks to aid in training, these networks are not needed in inference. In other words, our method is suitable for most MC3D-Det to learn perspective-invariant features. To test the effectiveness of our framework, BEVDepth (Li et al., 2023a) is instantiated as our main pipeline. The original detection loss $\mathcal{L}_{det}$ of BEVDepth is used as the main 3D detection supervision on the source domain, and depth supervision of BEVDepth has been replaced by $\mathcal{L}_{pg}$. In summary, our final loss of our work is:

$$\mathcal{L} = \lambda_s \mathcal{L}_{det} + \lambda_s \mathcal{L}_{render} + \lambda_s \mathcal{L}_{pg} + \lambda_s \mathcal{L}_{ps} + \lambda_t \mathcal{L}_{con}, \tag{8}$$

where $\lambda_s$ sets to 1 for source domain and sets to 0 for target domain, and the opposite is $\lambda_t$. In other words, $\mathcal{L}_{con}$ is not used under the DG protocol.

## 5 EXPERIMENT

To verify the effectiveness, we elaborately use both DG and UDA protocol for MC3D-Det. The details of datasets, evaluation metrics and implementation refer to appendix B.

### 5.1 DOMAIN GENERALIZATION BENCHMARK

For DG protocol, we replicate and compare the DG-BEV (Wang et al., 2023a) and the baseline BEVDepth (Li et al., 2023a). As shown in Tab. 1, our method has achieved significant improvement in the target domain. It demonstrates that IFV as a bridge can help learn perspective-invariant features against domain shifts. In addition, our approach does not sacrifice performance in the source domain and even has some improvement in most cases. It is worth mentioning that DeepAccident was collected from a Carla virtual engine, and our algorithm also achieved satisfactory generalization ability by training on DeepAccident. In addition, we have tested other MC3D-Det methods, and their generalization performance is very poor without special design as shown in Sec. 5.2.

### 5.2 UNSUPERVISED DOMAIN ADAPTATION BENCHMARK

To further validate debiasing on target domain, we also established a UDA benchmark and applied UDA methods (including Pseudo Label, Coral (Sun & Saenko, 2016), and AD (Ganin & Lempitsky,

| Nus → Lyft | | Source Domain (nuScenes) | | | | | Target Domain (Lyft) | | | | |
|---|---|---|---|---|---|---|---|---|---|---|---|
| Method | Target-Free | mAP↑ | mATE↓ | mASE↓ | mAOE↓ | NDS↑ | mAP↑ | mATE↓ | mASE↓ | mAOE↓ | NDS* ↑ |
| Oracle | | - | - | - | - | - | 0.598 | 0.474 | 0.152 | 0.092 | 0.679 |
| BEVDepth | ✓ | 0.326 | 0.689 | 0.274 | 0.581 | 0.395 | 0.114 | 0.981 | 0.174 | 0.413 | 0.296 |
| DG-BEV | ✓ | 0.330 | 0.692 | **0.272** | 0.584 | 0.397 | 0.284 | 0.768 | 0.171 | 0.302 | 0.435 |
| **PD-BEV** | ✓ | **0.334** | **0.688** | 0.276 | **0.579** | **0.399** | **0.304** | **0.709** | **0.169** | **0.289** | **0.458** |
| Pseudo Label | | 0.320 | 0.694 | 0.276 | 0.598 | 0.388 | 0.294 | 0.743 | 0.172 | 0.304 | 0.443 |
| Coral | | 0.318 | 0.696 | 0.283 | 0.592 | 0.387 | 0.281 | 0.768 | 0.174 | 0.291 | 0.435 |
| AD | | 0.312 | 0.703 | 0.288 | 0.596 | 0.381 | 0.277 | 0.771 | 0.174 | 0.288 | 0.381 |
| **PD-BEV**[+] | | **0.331** | **0.686** | **0.275** | **0.591** | **0.396** | **0.316** | **0.684** | **0.165** | **0.241** | **0.476** |
| Lyft → Nus | | Source Domain (Lyft) | | | | | Target Domain (nuScenes) | | | | |
| Method | Target-Free | mAP↑ | mATE↓ | mASE↓ | mAOE↓ | NDS*↑ | mAP↑ | mATE↓ | mASE↓ | mAOE↓ | NDS*↑ |
| Oracle | | - | - | - | - | - | 0.516 | 0.551 | 0.163 | 0.169 | 0.611 |
| BEVDepth | ✓ | **0.598** | **0.474** | 0.152 | 0.092 | **0.679** | 0.098 | 1.134 | 0.234 | 1.189 | 0.176 |
| DG-BEV | ✓ | 0.591 | 0.491 | 0.154 | 0.092 | 0.672 | 0.251 | 0.751 | 0.202 | 0.813 | 0.331 |
| **PD-BEV** | ✓ | 0.593 | 0.478 | **0.150** | **0.084** | 0.677 | **0.263** | **0.746** | **0.186** | **0.790** | **0.344** |
| Pseudo Label | | 0.580 | 0.538 | 0.153 | **0.079** | 0.657 | 0.261 | 0.744 | 0.201 | 0.819 | 0.306 |
| Coral | | 0.574 | 0.511 | 0.164 | 0.105 | 0.649 | 0.244 | 0.767 | 0.212 | 0.919 | 0.302 |
| AD | | 0.568 | 0.521 | 0.161 | 0.126 | 0.649 | 0.247 | 0.761 | 0.223 | 0.902 | 0.309 |
| **PD-BEV**[+] | | **0.589** | **0.489** | **0.150** | 0.091 | **0.672** | **0.280** | **0.733** | **0.182** | **0.776** | **0.358** |
| DeepAcci → Nus | | Source Domain (DeepAccident) | | | | | Target Domain (nuScenes) | | | | |
| Method | Target-Free | mAP↑ | mATE↓ | mASE↓ | mAOE↓ | NDS*↑ | mAP↑ | mATE↓ | mASE↓ | mAOE↓ | NDS*↑ |
| Oracle | | - | - | - | - | - | 0.516 | 0.551 | 0.163 | 0.169 | 0.611 |
| BEVDepth | ✓ | 0.334 | 0.517 | 0.741 | 0.274 | 0.412 | 0.087 | 1.100 | 0.246 | 1.364 | 0.169 |
| DG-BEV | ✓ | 0.331 | 0.519 | 0.757 | 0.264 | 0.408 | 0.159 | 1.075 | 0.232 | 1.153 | 0.207 |
| **PD-BEV** | ✓ | **0.345** | **0.499** | **0.735** | **0.251** | **0.425** | **0.187** | **0.931** | 0.229 | **0.967** | **0.239** |
| Pseudo Label | | 0.312 | 0.522 | 0.785 | 0.271 | 0.393 | 0.151 | 1.112 | 0.238 | 1.134 | 0.202 |
| Coral | | 0.314 | 0.544 | 0.796 | 0.274 | 0.388 | 0.164 | 1.045 | 0.242 | 1.104 | 0.208 |
| AD | | 0.312 | 0.539 | 0.787 | 0.263 | 0.391 | 0.166 | 1.013 | 0.251 | 1.073 | 0.207 |
| **PD-BEV**[+] | | **0.344** | **0.488** | **0.737** | **0.248** | **0.426** | **0.207** | **0.862** | **0.235** | **0.962** | **0.260** |
| DeepAcci → Nus | | Source Domain (DeepAccident) | | | | | Target Domain (Lyft) | | | | |
| Method | Target-Free | mAP↑ | mATE↓ | mASE↓ | mAOE↓ | NDS*↑ | mAP↑ | mATE↓ | mASE↓ | mAOE↓ | NDS*↑ |
| Oracle | | - | - | - | - | - | 0.598 | 0.474 | 0.152 | 0.092 | 0.679 |
| BEVDepth | ✓ | 0.334 | 0.517 | 0.741 | 0.274 | 0.412 | 0.045 | 1.219 | 0.251 | 1.406 | 0.147 |
| DG-BEV | ✓ | 0.331 | 0.519 | 0.757 | 0.264 | 0.408 | 0.135 | 1.033 | 0.269 | 1.259 | 0.189 |
| **PD-BEV** | ✓ | **0.345** | **0.499** | **0.735** | **0.251** | **0.425** | **0.151** | **0.941** | **0.242** | **1.130** | **0.212** |
| Pseudo Label | | 0.323 | 0.531 | 0.768 | 0.271 | 0.399 | 0.132 | 1.113 | 0.281 | 1.241 | 0.185 |
| Coral | | 0.308 | 0.573 | 0.797 | 0.284 | 0.378 | 0.145 | 1.004 | 0.254 | 1.129 | 0.196 |
| AD | | 0.304 | 0.554 | 0.796 | 0.274 | 0.381 | 0.148 | 0.997 | 0.262 | 1.189 | 0.197 |
| **PD-BEV**[+] | | **0.330** | **0.517** | **0.737** | **0.240** | **0.416** | **0.171** | **0.871** | **0.212** | **1.043** | **0.238** |

Table 1: Comparison of different approaches on DG and UDA protocols. Target-Free means DG protocol. Pseudo Label, Coral, and AD are applied in DG-BEV on UDA protocol. Following (Wang et al., 2023a), nuScenes is evaluated according to the original NDS as source domain. Other results are evaluated only for the 'car' category by NDS*.

2015)) on DG-BEV. As shown in Tab. 1, our algorithm achieved significant performance improvement. This is mainly attributed to the perspective debiasing, which fully utilizes the 2D detector with better generalization performance to correct the spurious geometric information of 3D detector. Additionally, we found that most algorithms tend to degrade performance on the source domain, while our method is relatively gentle. It is worth mentioning that we found that AD and Coral show significant improvements when transferring from a virtual dataset to a real dataset, but exhibit a decline in performance when testing on real-to-real testing. This is because these two algorithms are designed to address style changes, but in scenarios with small style changes, they may disrupt semantic information. As for the Pseudo Label algorithm, it can improve the model's generalization performance by increasing confidence in some relatively good target domains, but blindly increasing confidence in target domains can actually make the model worse.

## 5.3 ABLATION STUDY

To further demonstrate the effectiveness of our proposed algorithm, we conducted ablation experiments on three key components: 2D information injection $\mathcal{L}_{ps}$ (DII), source domain debiasing $\mathcal{L}_{render}$ (SDB), and target domain debiasing $\mathcal{L}_{con}$ (TDB). DII and MVSR are designed for the source domain, while TDB is designed for the target domain. In other words, we report the re-

| DII | SDB | TDB | Nus → Lyft | | DeepAcci → Lyft | |
|---|---|---|---|---|---|---|
| | | | mAP ↑ | NDS*↑ | mAP↑ | NDS*↑ |
| | | | 0.279 | 0.433 | 0.132 | 0.188 |
| ✓ | | | 0.290 | 0.438 | 0.143 | 0.205 |
| | ✓ | | 0.300 | 0.453 | 0.147 | 0.209 |
| ✓ | ✓ | | 0.304 | 0.458 | 0.151 | 0.212 |
| ✓ | ✓ | ✓ | **0.316** | **0.476** | **0.171** | **0.238** |

Table 2: Ablation study of different modules of PD-BEV. 2D information injection (DII), source domain debiasing (SDB), and target domain debiasing (TDB). TDB is only used for UDA protocol. In other words, the bottom line is the UDA result, and the rest is the DG result.

| Nus → Lyft | w/o ours | | w ours | |
|---|---|---|---|---|
| | mAP ↑ | NDS*↑ | mAP↑ | NDS*↑ |
| BEVDet | 0.104 | 0.275 | 0.296 | 0.446 |
| BEVFormer | 0.084 | 0.246 | 0.208 | 0.355 |
| FB-OCC | 0.113 | 0.294 | 0.301 | 0.454 |

Table 3: The plug-and-play capability testing of our method. We tested more MC3D-Det algorithms under the DG and tried to add our algorithm for further improvement.

sults under the UDA protocol only when using TDB, while the results of other components are reported under the DG protocol. As presented in Tab 2, each component has yielded improvements, with SDB and TDB exhibiting relatively significant relative improvements. SDB can better capture perspective-invariant and more generalizable features, while TDB leverages the strong generalization ability of 2D to facilitate the correction of spurious geometric features of the 3D detector in the target domain. DII makes the network learn more robust features by adding supervision to the image features in advance. These findings underscore the importance of each component in our algorithm and highlight the potential of our approach for addressing the challenges of domain gap in MC3D-Det.

## 5.4 FURTHER DISCUSSION

Here we try to migrate our framework to more MC3D-Det methods to prove the universality capability. We also give some visualizations to demonstrate the effectiveness of our framework.

**The plug-and-play capability of the method.** Our framework is model-agnostic. Any MC3D-Det algorithm with image feature and BEV feature can be embedded with our algorithm. Our algorithm framework is applied to BEVDet (Huang et al., 2021), BEVformer (Li et al., 2022c) and FB-BEV (Li et al., 2023b), as shown in Sec. 5.2. As the results show, our method can significantly improve the performance of these algorithms. This is because our algorithm can help the network learn perspective-invariant features against domain shifts.

**Perspective Debiasing.** To better explain the effect of our

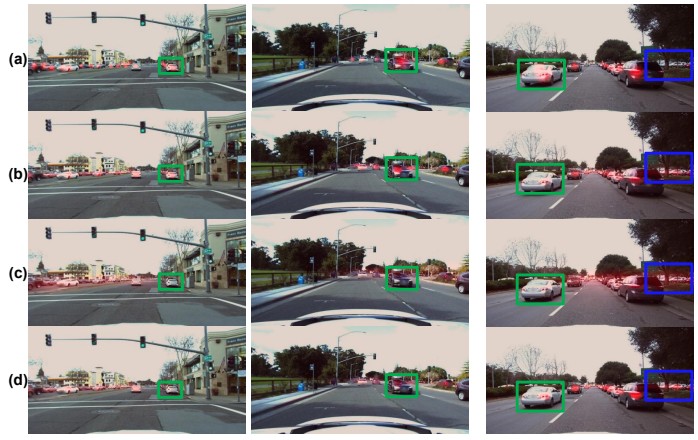

Figure 3: Visualization of heatmaps on target domain: (a) ground-truth, (b) 2D detector, (c) rendered from IVF, and (d) revised by 2D detector. The green rectangles indicates that our algorithm has improved the confidence of the detector prediction. The blue rectangles represent unlabeled objects that our algorithm detects. Please zoom in while maintaining the color.

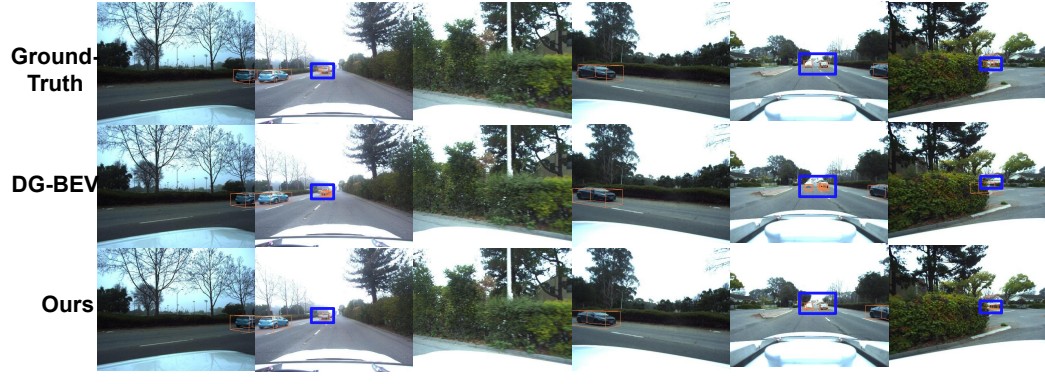

**Ground-Truth**

**DG-BEV**

**Ours**

**(a) Left-Front    (b) Front    (c) Right-Front  (d) Right-Back    (e) Back    (f) Left-Back**

Figure 4: Visualization of final MC3D-Det results. Our approach allows for more accurate detection and greatly reduces the presence of duplicate boxes. In front (b) and back (e) view, our method predicts more accurate and fewer duplicate boxes than DG-BEV. In left-back view, our method detects the location of the object more accurately. Please zoom in while maintaining the color.

perspective debiasing, we visualizes the heatmaps of 2D detector and the IFV in Fig. 3 (UDA protocol for Nus→Lyft). In the target domain, the 2D detector has good generalization performance and can accurately detect the center of the object in Fig. 3 (b). However, the heatmap rendered from the IFV is very spurious and it is very difficult to find the center of the objectin in Fig. 3 (c). Fig. 3 (d) shows that rendered heatmaps of IFV can be corrected effectively with 2D detectors.

**Visualization.** To better illustrate our algorithm, we visualized the final detection results of our algorithm and DG-BEV. As shown in Fig. 4, the detection results of our algorithm are more accurate, especially for the detection of distant objects. And our algorithm has fewer duplicate boxes, because the 2D detector can effectively correct the spurious geometric feature of the 3D detector and improve the confidence. We further visualized some interesting cases as shown in Fig. 5, and our algorithm can even detect some results that were not labeled in the original dataset, because the 2D detector has more generalization performance and further improves the performance of the 3D detector.

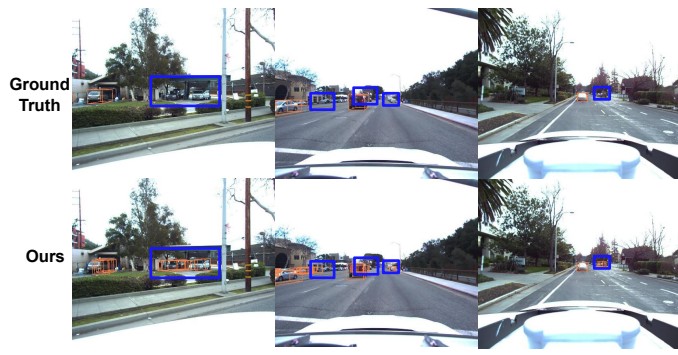

**Ground Truth**

**Ours**

Figure 5: Detected unlabeled objects. The first line is the 3D box of the ground-truth, and the second line is the detection result predicted by our algorithm. The blue box indicates that our algorithm can detect some unlabeled boxes. Please zoom in while maintaining the color.

## 6  SUMMARY

This paper proposes a framework for multi-camera 3D object detection (MC3D-Det) based on perspective debiasing to address the issue of poor generalization for unseen domains. We firstly render the semantic maps of different view from BEV features. We then use 3D boxes or pre-trained 2D detector to correct the spurious BEV features. Our framework is model-agnostic, and we demonstrate its effectiveness by optimizing multiple MC3D-Det methods. Our algorithms have achieved significant improvements in both DG and UDA protocols. Additionally, we explored training only on virtual annotations to achieve real-world MC3D-Det tasks.

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

## A    THE DERIVATION DETAILS OF PERSPECTIVE BIAS

To detect the object's location $L = [x, y, z]$ at the BEV space, corresponding to the image plane $[u, v]$, most MC3D-Det method involves two essential steps: (1) get the the image features from the j-th camera by the image encoder $F_{img}$. (2) map these feature into BEV space and fuse them to get the final location of objects by BEV encoder $F_{bev}$:

$$
\begin{aligned}
L &= F_{bev}(F_{img}(I_1), ..., F_{img}(I_N), K, E) \\
&= L_{gt} + \Delta L_{img} + \Delta L_{bev},
\end{aligned}
\tag{9}
$$

where $L_{gt}$, $\Delta L_{img}$ and $\Delta L_{bev}$ are the ground-truth depth and the bias of img encoder ($F_{img}$) and BEV encoder ($F_{bev}$). Before BEV feature fusion, object's depth $d(u, v)_{img} = d(u, v)_{gt} + \Delta L(u, v)_{img}$ needs to be extracted with image encoder, which will cause the domain bias of image encoder $\Delta L(u, v)_{img}$. Based on the estimated depth $d(u, v)_{img}$, the object can be lifted to the BEV space $[x', y', z']$:

$$
[x', y', z'] = E^{-1} K^{-1} [ud(u, v)_{img}, vd(u, v)_{img}, d(u, v)_{img}],
\tag{10}
$$

where $K$ and $E$ represent the camera intrinsic and camera extrinsic:

$$
K = \begin{bmatrix} f & 0 & c_u \\ 0 & f & c_v \\ 0 & 0 & 1 \end{bmatrix}, K^{-1} = \begin{bmatrix} 1/f & 0 & -c_u/f \\ 0 & 1/f & -c_v/f \\ 0 & 0 & 1 \end{bmatrix}
$$
$$
E = \begin{bmatrix} \cos\theta & 0 & \sin\theta & \big| & -t_x \\ 0 & 1 & 0 & \big| & -t_z \\ -\sin\theta & 0 & \cos\theta & \big| & -t_z \end{bmatrix}, E^{-1} = \begin{bmatrix} \cos\theta & 0 & -\sin\theta & \big| & t_x \\ 0 & 1 & 0 & \big| & t_z \\ \sin\theta & 0 & \cos\theta & \big| & t_z \end{bmatrix}.
\tag{11}
$$

It is worth noting that the camera extrinsic $E$ is simplified. We assume that the camera and ego coordinate systems are always in the same horizontal plane. Then, the special coordinate $[x', y', z']$ is:

$$
\begin{aligned}
[x', y', z'] &= E^{-1} K^{-1} ud(u, v)_{img}, vd(u, v)_{img}, d(u, v)_{img} \\
&= E[\frac{d(u, v)_{img}(u - c_u)}{f}, \frac{d(u, v)_{img}(v - c_v)}{f}, d(u, v)_{img}] \\
&= [\frac{d(u, v)_{img}(u - c_u)\cos\theta}{f} - d(u, v)_{img}\sin\theta + t_x, \frac{d(u, v)_{img}(v - c_v)}{f} + t_y, \\
&\quad \frac{d(u, v)_{img}(u - c_u)\sin\theta}{f} + d(u, v)_{img}\cos\theta + t_z],
\end{aligned}
\tag{12}
$$

where $d(u, v)_{img}$ indicates the depth predicted by the image encoder. It is worth mentioning that this mapping process can be explicit or implicit, but they are all based on the depth information learned from the single image. After this, the BEV encoder further merges and modifies these to the final coordinate $[x, y, z]$:

$$
\begin{aligned}
[x', y', z'] &= [x, y, z] + \Delta L_{bev}(x, y, z) \\
&= [\frac{d(u, v)_{img}(u - c_u)\cos\theta}{f} - d(u, v)_{img}\sin\theta + t_x + \Delta L_{bev}(x|x, y, z), \\
&\quad \frac{d(u, v)_{img}(v - c_v)}{f} + t_y + \Delta L_{bev}(y|x, y, z), \\
&\quad \frac{d(u, v)_{img}(u - c_u)\sin\theta}{f} + d(u, v)_{img}\cos\theta + t_z + \Delta L_{bev}(z|x, y, z)],
\end{aligned}
\tag{13}
$$

where $\Delta L_{bev}(x|x, y, z)$ means that the domain bias of BEV encoder affects the change in the $x$ dimension. In order to quantitatively describe how domain bias manifests in a single perspective,

this object are re-projected to the image homogeneous coordinates with $d(u,v)_{img}$, $K$ and $E$:

$$
\begin{aligned}
[u'd_f, v'd_f, d_f] &= KE[x', y', z'] \\
&= K[\frac{d(u,v)_{img}(u-c_u)}{f} + \Delta L_{bev}(x|x,y,z)\cos(\theta) + \Delta L_{bev}(z|x,y,z)\sin(\theta), \\
&\quad \frac{d(u,v)_{img}(v-c_v)}{f} + \Delta L_{bev}(y|x,y,z), \\
&\quad d(u,v)_{img} + \Delta L_{bev}(z|x,y,z)\cos(\theta) - \Delta L_{bev}(x|x,y,z)\sin(\theta)] \\
&= [d(u,v)_{img}u + \Delta L_{bev}(x|x,y,z)(f\cos(\theta) - c_u\sin(\theta)) \\
&\quad + \Delta L_{bev}(z|x,y,z)(f\sin(\theta) + c_u\cos(\theta)), \\
&\quad d(u,v)_{img}v + f\Delta L_{bev}(y|x,y,z) + c_v\Delta L_{bev}(z|x,y,z)\cos(\theta) \\
&\quad - c_v\Delta L_{bev}(x|x,y,z)\sin(\theta), \\
&\quad d(u,v)_{img} + \Delta L_{bev}(z|x,y,z)\cos(\theta) - \Delta L_{bev}(x|x,y,z)\sin(\theta)].
\end{aligned}
\tag{14}
$$

Then we can calculate individual perspective bias $[\Delta u, \Delta v] = [u' - u, v' - v]$. To simplify the process, let's first calculate:

$$
\begin{aligned}
[\Delta u d_f, \Delta v d_f, d_f] &= [(u'-u)d_f, (v'-v)d_f, d_f] \\
&= [u'd_f - ud_f, v'd_f - vd_f, d_f] \\
&= [\Delta L_{bev}(x|x,y,z)(f\cos(\theta) + u\sin(\theta) - c_u\sin(\theta)) \\
&\quad + \Delta L_{bev}(z|x,y,z)(f\sin(\theta) + c_u\cos(\theta) - u\cos(\theta)), \\
&\quad f\Delta L_{bev}(y|x,y,z) + (c_v - v)(\Delta L_{bev}(z|x,y,z)\cos(\theta) \\
&\quad - \Delta L_{bev}(x|x,y,z)\sin(\theta)), \\
&\quad d(u,v)_{img} + \Delta L_{bev}(z|x,y,z)\cos(\theta) - \Delta L_{bev}(x|x,y,z)\sin(\theta)].
\end{aligned}
\tag{15}
$$

Then, we can calculate the structure of the perspective bias $[\Delta u, \Delta v]$ in the camera plane:

$$
\begin{aligned}
[\Delta u, \Delta v] &= [\frac{\Delta L_{bev}(x|x,y,z)(f + (u-c_u)\tan(\theta)) + \Delta L_{bev}(z|x,y,z)(f\tan(\theta) + c_u - u)}{d(u,v)_{img}secx(\theta) + \Delta L_{bev}(z|x,y,z) - \Delta L_{bev}(x|x,y,z)\tan(\theta)}, \\
&\quad \frac{\Delta L_{bev}(y|x,y,z)fsecx(\theta) + (\Delta L_{bev}(z|x,y,z) - \Delta L_{bev}(x|x,y,z)\tan(\theta))(c_v - v)}{d(u,v)_{img}secx(\theta) + \Delta L_{bev}(z|x,y,z) - \Delta L_{bev}(x|x,y,z)\tan(\theta)} \\
&= [\frac{(u-c_u)(\Delta L_{bev}(x|x,y,z)\tan(\theta) - \Delta L_{bev}(z|x,y,z))}{d(u,v)_{img}secx(\theta) + \Delta L_{bev}(z|x,y,z) - \Delta L_{bev}(x|x,y,z)\tan(\theta)} \\
&\quad + \frac{\Delta L_{bev}(x|x,y,z)f + \Delta L_{bev}(z|x,y,z)f\tan(\theta)}{d(u,v)_{img}secx(\theta) + \Delta L_{bev}(z|x,y,z) - \Delta L_{bev}(x|x,y,z)\tan(\theta)}, \\
&\quad \frac{(v-c_v)(\Delta L_{bev}(x|x,y,z)\tan(\theta) - \Delta L_{bev}(z|x,y,z)) + \Delta L_{bev}(y|x,y,z)fsecx(\theta)}{d(u,v)_{img}secx(\theta) + \Delta L_{bev}(z|x,y,z) - \Delta L_{bev}(x|x,y,z)\tan(\theta)}] \\
&= [\frac{k_u(u-c_u) + b_u}{d(u,v)}, \frac{k_v(v-c_v) + b_u}{d(u,v)}].
\end{aligned}
\tag{16}
$$

This final result is too complicated for us to analyze the problem, so we bring in $d(u,v)_{img} = d(u,v)_{gt} + \Delta L(u,v)_{img}$ and simplify further:

$$
\begin{aligned}
[\Delta u, \Delta v] &= [\frac{(u-c_u)(\Delta L_{bev}(x|x,y,z)\tan(\theta) - \Delta L_{bev}(z|x,y,z))}{(d(u,v)_{gt} + \Delta L(u,v)_{img})secx(\theta) + \Delta L_{bev}(z|x,y,z) - \Delta L_{bev}(x|x,y,z)\tan(\theta)} \\
&\quad + \frac{\Delta L_{bev}(x|x,y,z)f + \Delta L_{bev}(z|x,y,z)f\tan(\theta)}{(d(u,v)_{gt} + \Delta L(u,v)_{img})secx(\theta) + \Delta L_{bev}(z|x,y,z) - \Delta L_{bev}(x|x,y,z)\tan(\theta)}, \\
&\quad \frac{(v-c_v)(\Delta L_{bev}(x|x,y,z)\tan(\theta) - \Delta L_{bev}(z|x,y,z)) + \Delta L_{bev}(y|x,y,z)fsecx(\theta)}{(d(u,v)_{gt} + \Delta L(u,v)_{img})secx(\theta) + \Delta L_{bev}(z|x,y,z) - \Delta L_{bev}(x|x,y,z)\tan(\theta)}] \\
&= [\frac{k_u(u-c_u) + b_u}{d(u,v)}, \frac{k_v(v-c_v) + b_v}{d(u,v)}].
\end{aligned}
\tag{17}
$$

where $d(u,v) = (d(u,v)_{gt} + \Delta L(u,v)_{img})secx(\theta) + \Delta L_{bev}(z|x,y,z) - \Delta L_{bev}(x|x,y,z)\tan(\theta)$ represents the predicted depth on $[u,v]$ by the final model, and $k_u, b_u, k_v$, and, $b_u$ is related to the domain bias of BEV encoder$\Delta L_{BEV}$. Specifically, $k_u = \Delta L_{bev}(x|x,y,z)\tan(\theta) - \Delta L_{bev}(z|x,y,z)$, $b_u = \Delta L_{bev}(x|x,y,z)f + \Delta L_{bev}(z|x,y,z)f\tan(\theta)$, $k_v = \Delta L_{bev}(x|x,y,z)\tan(\theta) - \Delta L_{bev}(z|x,y,z)$, $b_v = \Delta L_{bev}(y|x,y,z)fsecx(\theta)$.

Here, let's reanalyze the causes of the model bias and give us why our method works:

• The bias of image encoder $\Delta d(u,v)_{img}$ is caused by overfitting to the camera intrinsic parameters and limited viewpoint. The camera intrinsic affects the depth estimation of the network, which can be solved by estimation the virtual normalization depth as elaborated in Sec. 4.2.2. Limited viewpoint refers to the fact that the network will reason based on the relationship between the vehicle, the road surface and the background. It is worth mentioning that the viewpoint is difficult to decouple, because it is difficult to get the image of different views in the real world. However, on cross-domain testing, both the camera intrinsic parameters and viewpoint will change dramatically, which will lead to greater bias for image encoder.

• The bias of BEV encoder $\Delta d(u,v)_{bev}$ is indirectly affected by the bias of image encoder. The changes of viewpoint and environment will lead to the distribution shift of foreground and background features, which causes degradation of the model. The BEV encoder is to modify the geometry information of the image encoder. Therefore, $\Delta d(u,v)_{img}$ will further affect $\Delta d(u,v)_{bev}$.

In summary, the algorithms used often have a tendency to overfit to the viewpoint, camera parameters and change, caused by limited real data. Essentially, our aim is to ensure that the features extracted by the network from different perspectives remain consistent against different viewpoint, camera parameters and environment. Specifically, we hope that the network can learn perspective-independent features in the image encoder. Additionally, we expect the instance features to remain consistent even when observed from different angles or fused with the surrounding environment in different ways. For the BEV encoder, we anticipate that the instance features will be generalized when fused in different BEV spatial distributions. These improvements will lead to a more robust and accurate model.

# B  DATASETS AND IMPLEMENTATION

## B.1  DATASETS.

We evaluate our proposed approach on three datasets, including one virtual and two real autonomous driving datasets: DeepAccident (Wang et al., 2023b), nuScenes (Caesar et al., 2020), and Lyft (Kesten et al., 2019). Different datasets have different cameras with different intrinsic parameters and environment as shown in Tab. 4.

| Dataset | Location | Shape | Focal Length |
|---------|----------|-------|--------------|
| DeepAccident | Virtual engine | (900,1600) | 1142,560 |
| nuScenes | Boston,SG. | (900,1600) | 1260 |
| Lyft | Palo Alto | (1024, 1224),(900,1600) | 1109,878 |

Table 4: Dataset Overview. SG: Singapore.

In order to gain a deeper understanding of the domain shifts the various datasets, we conducted a separate visualization of each of the three datasets. Specifically, Fig. 6 depicts the DeepAccident dataset, while Fig. 7 pertains to the nuScenes dataset, and Fig. 8 illustrates the Lyft dataset.

## B.2  EVALUATION METRICS.

Following (Wang et al., 2023a), we adopt NDS* as a substitute metric for assessing the generalization performance of our model. NDS* is an extension of the official NDS metric used in nuScenes. Since the attribute labels and velocity labels are not directly comparable in different datasets, NDS* excludes the mean Average Attribute Error (mAAE) and the mean Average Velocity Error (mAVE) from its computation. Instead, it focuses on evaluating the mean Average Precision (mAP), the mean

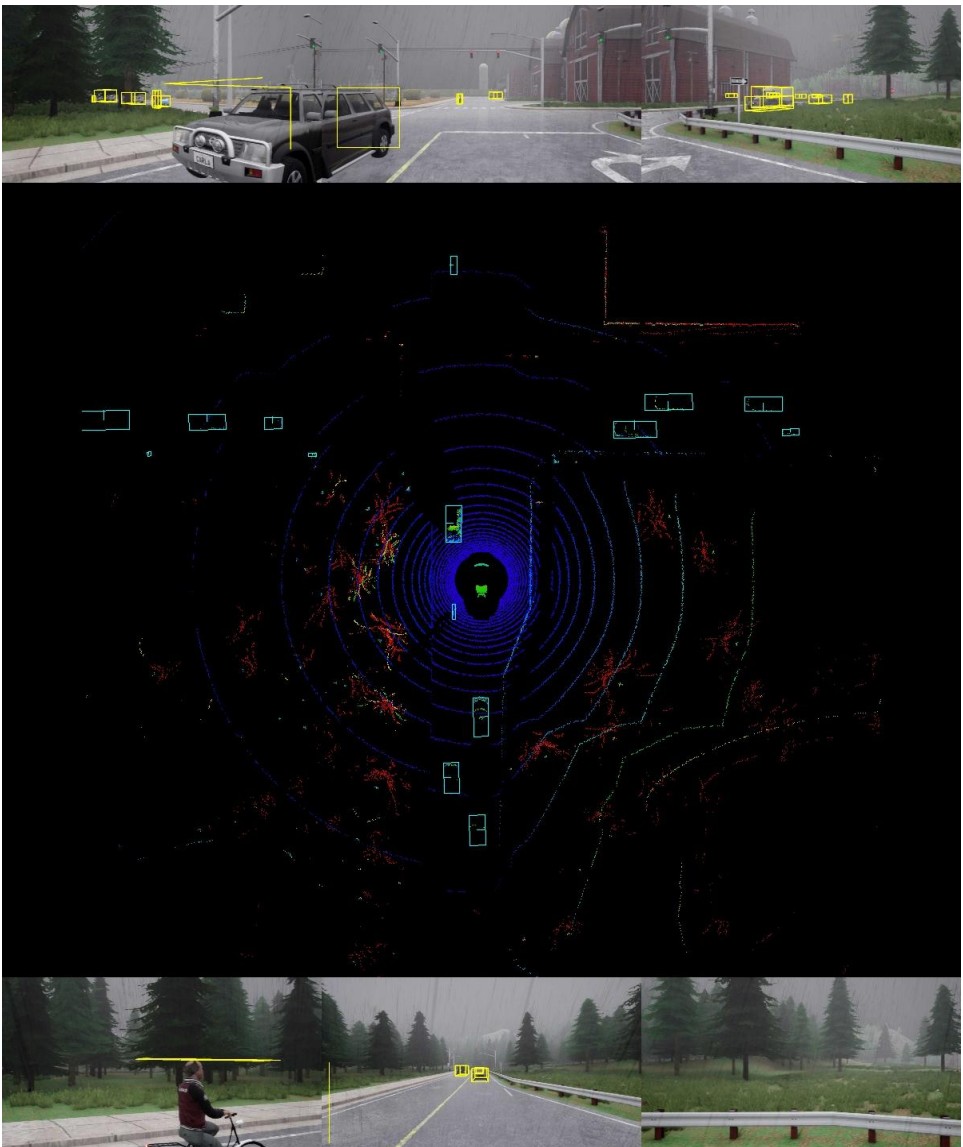

Figure 6: Visualization of DeepAccident. Please zoom in while maintaining the color.

Average Translation Error (mATE), the mean Average Scale Error (mASE), and the mean Average Orientation Error (mAOE) to measure the model's performance:

$$\text{NDS}^* = \frac{1}{6}[3\,\text{mAP} + \sum_{\text{mTP}\in\mathbb{TP}} (1 - \min(1, \text{mTP}))], \tag{18}$$

It is worth noting that all metrics are reported on the validation in the range [-50m, 50m] along the x-y axes.

### B.3    IMPLEMENTATION DETAILS.

To validate the effectiveness of our proposed method, we use BEVDepth (Li et al., 2023a) as our baseline model. Following the training protocol in (Huang et al., 2021), we train our models using the AdamW (Loshchilov & Hutter, 2018) optimizer, with a gradient clip and a learning rate of 2e-4. We use a total batch size of 32 on 8 Tesla 3090 GPUs. The final image input resolution for all datasets is set to $704 \times 384$. We apply data augmentation techniques such as random flipping, random scaling and random rotation within a range of $[-5.4°, 5.4°]$ to the original images during training. It is worth

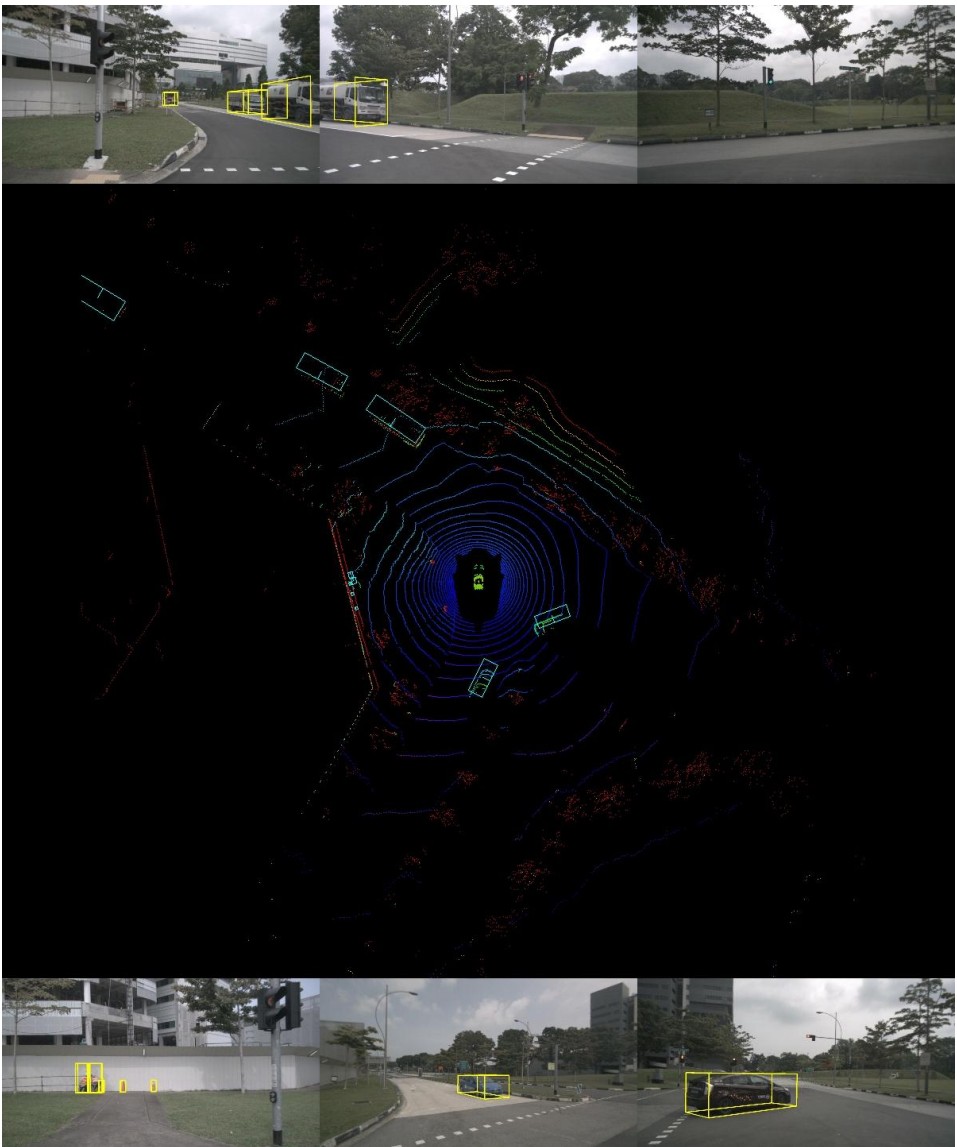

Figure 7: Visualization of nuScenes. Please zoom in while maintaining the color.

noting that the random scaling is with a range of $[-0.04, +0.11]$ for nuScenes and DeepAccident, and $[-0.04, +0.30]$ for Lyft, which is to better fit the different intrinsic parameters of camera on different datasets.

### B.4 NETWORK STRUCTURE

The new networks introduced in our paper include 2D Detector, BEV decoder and RenderNet. The network structure of BEVDepth has not changed at all.

• **2D Detector** is composed of a three-layer 2D convolution layer and center head detector (Yin et al., 2021). It takes as input the image features from different cameras and outputs 2D test results, which are supervised by $\mathcal{L}_{ps}$.

• **BEV decoder** consists of six layers of 2D convolution with a kernel size of 3, and its feature channel is 80, except for the last layer, which has 84 channels. It takes as input the BEV features from the BEV encoder and outputs intermediate feature $F'_{bev} \in \mathbb{R}^{C \times 1 \times X \times Y}$ and $F_{height} \in \mathbb{R}^{1 \times Z \times X \times Y}$ stated in Sec. 4.1, which is indirectly supervised by $\mathcal{L}_{render}$ and $\mathcal{L}_{con}$.

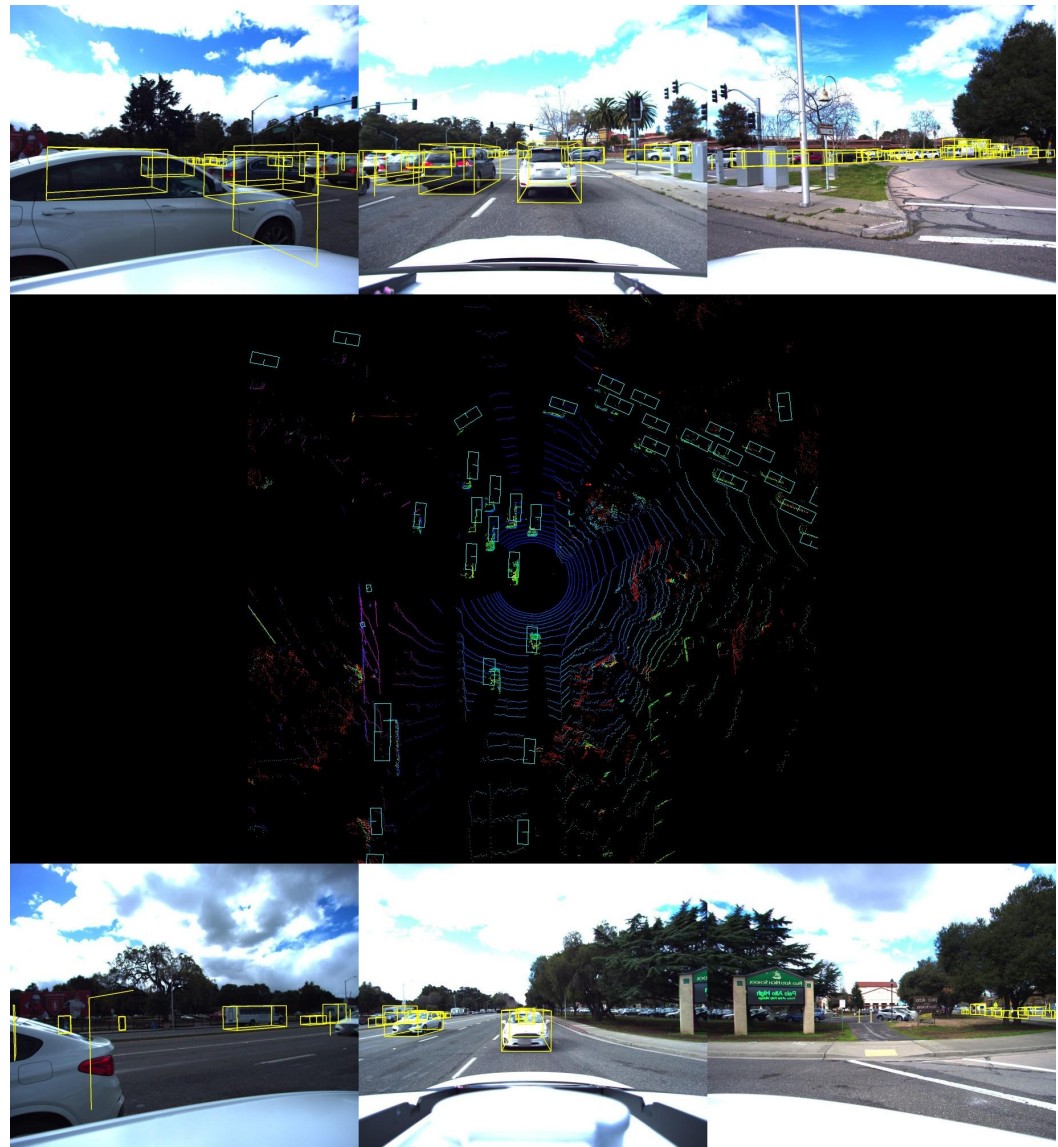

Figure 8: Visualization of Lyft. Please zoom in while maintaining the color.

• **RenderNet** consists of four layers of 2D convolution with a kernel size of 3, and its feature channel is $[256, 256, 128, 7 * N_{cls}]$. Here, $N_{cls}$ represents the number of classes, and each class has a channel classification header. The remaining six channels respectively represent the position bias of each dimension and the dimension (length, width, and height) of the object.

## C  DISCUSSION

### C.1  DISCUSSION OF SEMANTIC RENDERING

Semantic rendering has many hyperparameters, which we will discuss here: Perturbed range of pitch, yaw, roll, and the height of IVF are discussed as shown in Fig. 9. The values of these four parameters in the paper are 0.04,0.2,0.04 and 4, respectively. Based on this, we make changes to each parameter separately to see how they affect the final result. Based on Fig. 9, We found that the effect of the range of yaw $\Delta\theta_{yaw}$ and the height of IVF $Z$ is relatively large. It also shows that it is effective to synthesize new perspectives from different viewpoints.

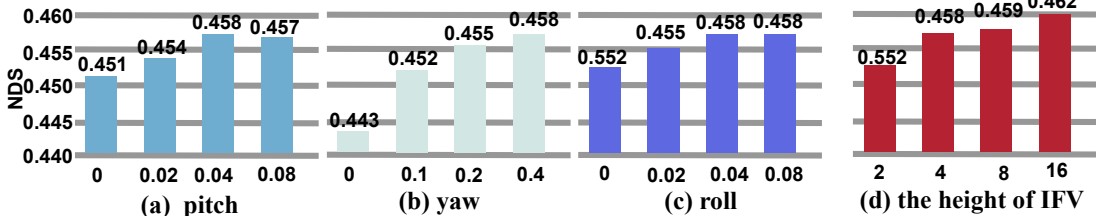

Figure 9: Ablation study of semantic rendering. Perturbed range of pitch $\Delta\theta_{pitch}$, yaw $\Delta\theta_{yaw}$, roll $\Delta\theta_{roll}$ and the height of IVF are discussed.

Furthermore, we evaluate the perturbed range of translation (X, Y, Z), and the results are presented in the table above.

| Perturbed X range (m) | mAP | NDS |
|---|---|---|
| $[0.0, 0.0]$ | 0.290 | 0.438 |
| $[-1.0, 1.0]$ | 0.295 | 0.446 |
| $[-2.0, 2.0]$ | 0.301 | 0.453 |
| $[-4.0, 4.0]$ | 0.294 | 0.443 |

Table 5: Ablation study of perturbed X.

| Perturbed X range (m) | mAP | NDS |
|---|---|---|
| $[0.0, 0.0]$ | 0.290 — | 0.438 |
| $[-1.0, 1.0]$ | 0.296 — | 0.444 |
| $[-2.0, 2.0]$ | 0.304 — | 0.457 |
| $[-4.0, 4.0]$ | 0.300 — | 0.449 |

Table 6: Ablation study of perturbed X.

| Perturbed X range (m) | mAP | NDS |
|---|---|---|
| $[0.0, 0.0]$ | 0.290 | 0.438 |
| $[-1.0, 1.0]$ | 0.292 | 0.441 |
| $[-2.0, 2.0]$ | 0.294 | 0.443 |
| $[-4.0, 4.0]$ | 0.284 | 0.422 |

Table 7: Ablation study of perturbed X.

The ablation in the table reveals that random translation of the observation position to render a new perspective can effectively enhance the model's performance. Notably, the difference between the camera extrinsic parameters of different datasets primarily pertains to the camera position and camera yaw angle. Among them, the camera position relative to the ego coordinate of car does not exceed 2m. Based on the findings presented in Figure 9 and the table above, it can be inferred that our proposed algorithm can significantly enhance the robustness of the model. In other words, our perturbation range already includes the camera extrinsic range of the target domain.

### C.2 COMPARISON OF DIFFERENT DEPTHS OF SUPERVISION

To further illustrate why we use the depth of the center of the foreground object (3D boxes provided) instead of the surface depth (LiDAR provided). We modified our algorithm with using different depth supervision, and it is worth mentioning that both monitors are converted to virtual depth by Eq. 6 for fair comparison. As shown in Tab. 8, under sim2real, no depth supervision is even better than LiDAR depth supervision. This is because the depth information provided by the virtual engine is the object surface, which may project the foreground object of the target domain to the

| Deep supervision | | DeepAcci → Lyft | | Nus → Lyft | |
|---|---|---|---|---|---|
| LiDAR | Boxes | mAP↑ | NDS*↑ | mAP↑ | NDS*↑ |
| | | 0.162 | 0.230 | 0.307 | 0.461 |
| ✓ | | 0.148 | 0.212 | 0.312 | 0.469 |
| | ✓ | **0.171** | **0.238** | **0.316** | **0.476** |

Table 8: Comparison of different depth supervision for PD-BEV$^+$.

wrong location on the BEV plane. Without deep supervision, the model will rely more on the BEV Encoder to learn the relationships between objects and the result will be more robust. It is worth mentioning that the depth supervision of boxes always achieve clearly optimal results.

### C.3 THE COMPARISON WITH 2D-3D CONSISTENT METHOD

The key difference between our render-based method and existing 2D-3D consistency approaches lies in the following aspects:

Geometric feature correction with less semantic destruction. Our rendering approach enables accurate projection of 2D images to the exact location of the BEV, thereby allowing the network to modify the geometric position of features with minimal semantic destruction. In contrast, 2D-3D consistency constraints have a significant impact on the entire network and often result in the destruction of semantic information. To compare our approach with existing 2D-3D consistency methods, we selected two representative approaches, namely Hybrid-Det (Yang et al., 2022) and MVC-Det (Lian et al., 2022). Specifically, we report the unsupervised domain adaptation result (mAP/NDS) of these methods in nuScenes (source) and Lyft (target) datasets:

| Method | nuScenes (source) | | Lyft (target) | |
|---|---|---|---|---|
| | mAP | NDS | mAP | NDS |
| DG-BEV | 0.330 | 0.397 | 0.284 | 0.435 |
| MVC-Det | 0.314 | 0.374 | 0.288 | 0.437 |
| Hybrid-Det | 0.301 | 0.361 | 0.292 | 0.441 |
| Ours | 0.331 | 0.396 | 0.316 | 0.476 |

Table 9: Comparison of methods on nuScenes and Lyft datasets

As shown in the table above, our algorithm achieves excellent performance. Although MVC-Det and Hybrid-Det exhibit improved performance in the target domain, their results in the source domain have deteriorated significantly. This is because other 2D-3D consistency methods enable them to fine-tune the entire network with semantic destruction. In contrast, our algorithm modifies the geometric space position of the feature, which leads to relatively stable results in the source domain.

The unified method for any detection head. MC3D-Det offers multiple detection heads, including a combination of classification and regression (Centerpoint) and end-to-end (DETR) approaches. However, designing a uniform differentiable mechanism that satisfies 2D-3D consistency is challenging due to the diverse nature of these detection heads. For instance, in Centerpoint, the final prediction is obtained by combining the outputs of the classification head and an offset regression head. It is difficult to supervise the classification head in this case effectively. In contrast, our approach presents a unified solution through a rendering approach.

Significant improvement on both source and target domains. Our approach is highly effective in improving the performance of both the source and target domains. In the source domain, our approach utilizes rendering to generate new views with different camera parameters, enabling the network to learn perspective-independent features. In the target domain, our approach employs a more robust 2D detector to correct the 3D results by rendering. The other 2D-3D consistency methods do not have much effect in the source domain because 3D supervision is inherently more accurate and informative. However, our approach stands out by enabling the supervision of new perspectives through rendering.

## C.4   More experiments about MC3D-Det

We also conducted experiments on several methods without BEV representation (DETR3D, PETR, SparseBEV) on cross-domain results (from nuScenes to Lyft) as shown in the table below.

| Method | w/o ours mAP/NDS | w/ ours mAP/NDS |
|---|---|---|
| DETR3D* | 0.008 / 0.044 | 0.028 / 0.076 |
| PETR* | 0.012 / 0.051 | 0.032 / 0.091 |
| SparseBEV* | 0.016 / 0.059 | 0.038 / 0.097 |
| BEVFormer | 0.084 / 0.246 | 0.208 / 0.355 |
| BEVDet | 0.104 / 0.275 | 0.296 / 0.446 |
| FB-BEV | 0.113 / 0.294 | 0.301 / 0.454 |
| BEVDepth | 0.114 / 0.296 | 0.304 / 0.458 |

Table 10: Comparison of methods with and without ours

Methods with an asterisk (*) do not have BEV representation, while all others do. For these methods without BEV representation, we leverage the LSS mechanism in BEVDet to build additional BEV presentation so that our algorithm can be used to improve these methods. Notably, our algorithm is model-agnostic; in other words, our algorithm can be applied to any method with image features and BEV features, as shown in Fig. 2. According to the table, we draw a number of conclusions:

(1) Methods without BEV presentation perform very poorly across domains compared to other algorithms. This can be attributed to their tendency to overfit camera extrinsic parameters in a learning way. Conversely, methods with BEV representation solely employ camera extrinsic parameters to project 2D image features into 3D space through a physical modeling form. This approach is highly resilient to variations in camera extrinsic parameters, thereby increasing its robustness and reliability. In conclusion, the BEV representation can effectively establish the connection of different perspectives through physical modeling, as opposed to learning. This way enables the model to have superior cross-domain generalization.

(2) Our algorithm has the potential to enhance the performance of DETR3D, PETR, and SparesBEV algorithms by rendering the new viewpoints from an auxiliary BEV presentation. This is because the process of rerendering new perspectives compels the network to acquire a generalizable BEV representation. The generalizable BEV representation, in turn, encourages the network to learn more robust visual features that mitigate the impact of overfitting camera parameters.

