# OpenReview forum: "Towards Generalizable Multi-Camera 3D Object Detection via Perspective Debiasing"
_ICLR.cc/2024/Conference — Submitted to ICLR 2024_

### Official Review · Reviewer_kBUt · 2023-10-22

**Soundness:** 3 good
**Presentation:** 3 good
**Contribution:** 3 good
**Rating:** 6
**Confidence:** 5

**Summary:**

Drawing from this insight (ie., 2D detection in a single-view (camera plane) often has a stronger ability to generalize than multi-camera 3D object detection), this paper leverages the 2D view prior to better construct the consistency between cross domains. The proposed method achieves excellent results in both DG and UDA benchmarks.

**Strengths:**

- The proposed method is novel, and can not bring about any inference latency cost.
- The performance improvements of all sub-modules are significant.
- The paper is well-organized and well-written.

**Weaknesses:**

- It is better to analyze that the 2D feature is more suitable to deal with the DA or UDA problems, ie., some statistical analysis or specific documentary evidence.
- In Table 1, are the settings of BEVDepth and PC-BEV aligned?
- Many papers have shown that adding a 2D detection prediction task for the MC3D-Det series detectors will significantly boost the performance. I'm worried about how much of your current rise is coming from the 2D prediction of the detector. So you should scrupulously add an ablation to show the effect of 2D prediction (even other sub-modules) for source domain only.
- You should discuss the relative works of consistent learning and extra 2D prediction, i.e., [a,b] etc.
[a] Probabilistic and Geometric Depth: Detecting Objects in Perspective.
[b] Towards 3D Object Detection with 2D Supervision.

**Questions:**

Please see the Weaknesses.

---

> ### Author Response · Authors · 2023-11-21
> **The first response to reviewer tA1k**
>
> We sincerely appreciate the time and effort the reviewer dedicated to reviewing our paper and providing constructive comments. Below are our responses to the raised concerns.
>
> # **1. Suitability of 2D detector**
>
> Here, we re-elaborate the suitability of the 2D detector from four aspects, including statistical indexes, theory, ablation study, and related work:
>
>
> **Statistical result.** Comparing the performance of 2D and 3D predictions can be challenging due to the lack of suitable metrics. To address this issue, we adopt a straightforward approach by projecting the 3D prediction result center onto the 2D front camera and evaluating the Focal loss on the front camera. We report the average Focal loss in the source domain (nuScenes) and the target domain (Lyft) for both 2D and 3D predictions, as shown below:
>
>
> | Method| Source:nuScenes | Target:Lyft |
> | :---:  | :---:  |:---:  |
> | 2D | 0.6557 | 0.7492 |
> | 3D | 0.6623 | 1.1412 |
>
>
> The results demonstrate that 2D detection results are significantly more stable than 3D results in both the source and target domains, with the 2D detection results being notably higher than the 3D results in the target domain.
>
>
> **Experimental proof.** In Table 2, we conduct an ablation experiment on our approach by explicitly adding 2D extra supervision (training on the source domain) and rerendering, which leads to significant improvements.
>
>
>
> **The mathematical derivation.** We provide the mathematical derivation that the 3D prediction error caused by domain shift can be reflected in a single perspective (2D image plane), and the final error on the single perspective is defined as Eq. (2). The detailed proof is presented in Appendix C. This derivation demonstrates that correcting errors in different 2D planes can reduce the errors in the final 3D prediction results, providing a theoretical basis for our method.
>
>
>
> **Proof of related work:** Extensive research has been undertaken to exploit 2D techniques in the context of 3D applications. One such approach involves incorporating precise 2D outcomes as supplementary information into the network, thereby augmenting its capabilities [1, 2]. However, these methods tend to compromise the original network results and increase inference delays, whereas our proposed method preserves the original network structure. Another method entails utilizing 2D-3D consistency to enhance network detection capabilities [3, 4]. Although this approach bears some resemblance to our proposed method, we will elucidate the differences between our algorithm and these existing methods in part 4.
>
>
>
> [1] Wang Z, Huang Z, Fu J, et al. Object as Query: Lifting Any 2D Object Detector to 3D Detection. Proceedings of the IEEE/CVF International Conference on Computer Vision. 2023: 3791-3800.
>
> [2] Yang C, Chen Y, Tian H, et al. BEVFormer v2: Adapting Modern Image Backbones to Bird's-Eye-View Recognition via Perspective Supervision. Proceedings of the IEEE/CVF Conference on Computer Vision and Pattern Recognition. 2023: 17830-17839.
>
> [3] Yang J, Wang T, Ge Z, et al. Towards 3D Object Detection with 2D Supervision. arXiv preprint arXiv:2211.08287, 2022.
>
> [4] Lian Q, Xu Y, Yao W, et al. Semi-supervised monocular 3d object detection by multi-view consistency. European Conference on Computer Vision. Cham: Springer Nature Switzerland, 2022: 715-731.
>
>
> # **2. The fair comparison with BEVDepth**
>
>
> **The restatement of fairness in Table 1.** All parameters of the comparison in Table 2 are the same, including image resolution, learning rate and training rounds. The improvement of our algorithm is only the addition of additional auxiliary networks and auxiliary constraints.
>
>
> **The comparison with a stronger baseline.** To align the result with the original paper, we improve our baseline by changing the backbone to R101-DCN and the input resolution to 768 $\times$ 1408. Additionally, both our proposed method and DG-BEV modify the basic backbone and image resolution. These approaches are tested on a cross-dataset (from nuScenes to Lyft), and the results for the source domain (nuScenes) and the target domain (Lyft) are presented in the table below:
>
>
> | Method |Protocol| nuScenes (mAP / NDS) | Lyft (mAP / NDS) |
> | :---:  | :---:  |:---:  | :---:  |
> | BEVDepth| DG  |0.414 / 0.538 | 0.120 / 0.304 |
> | DG-BEV | DG  |0.409 / 0.531 | 0.279 / 0.431 |
> | Ours | DG  | 0.421 / 0.541 | 0.324 / 0.484 |
> | Ours |  UDA  | 0.417 / 0.539 | 0.335 / 0.497 |
>
> Here, DG represents domain generalization, and UDA represents unsupervised domain adaptation. As shown in the table, the results in the source domain (nuScenes) have significantly improved, but the improvement in the target domain (Lyft) is less pronounced. This indicates that increasing the model size and image resolution has a limited effect on the model's cross-domain performance. It is worth noting that our proposed method still significantly improves the results in the target domain.

---

> ### Author Response · Authors · 2023-11-21
> **The second response to reviewer tA1k**
>
> # **3. Ablation study**
>
> In Table 2, we conducted an ablation experiment. In particular, in our comparison, the addition of 2D supervision improved performance, and our rerendering (source
> domain debiasing) also improves performance.
>
>
>
> # **4. The comparison with 2D-3D consistent method**
>
> The key difference between our render-based method and existing 2D-3D consistency approaches lies in the following aspects:
>
>
> **Geometric feature correction with less semantic destruction.** Our rendering approach enables accurate projection of 2D images to the exact location of the BEV, thereby allowing the network to modify the geometric position of features with minimal semantic destruction. In contrast, 2D-3D consistency constraints have a significant impact on the entire network and often result in the destruction of semantic information. To compare our approach with existing 2D-3D consistency methods, we selected two representative approaches, namely Hybrid-Det [1] and MVC-Det[2]. Specifically, we report the unsupervised domain adaptation result (mAP/NDS) of these methods in nuScenes (source) and Lyft (target) datasets:
>
>
> | Method| nuScenes (source) | Lyft (target) |
> | :---:  | :---:  |:---:  |
> | DG-BEV |0.330  / 0.397 | 0.284 / 0.435 |
> | MVC-Det | 0.314 / 0.374 | 0.288 / 0.437 |
> | Hybrid-Det | 0.301 / 0.361 | 0.292 / 0.441 |
> | Ours | 0.331 / 0.396 | 0.316 / 0.476 |
>
>
> As shown in the table above, our algorithm achieves excellent performance. Although MVC-Det and Hybrid-Det exhibit improved performance in the target domain, their results in the source domain have deteriorated significantly. This is because other 2D-3D consistency methods enable them to fine-tune the entire network with semantic destruction. In contrast, our algorithm modifies the geometric space position of the feature, which leads to relatively stable results in the source domain.
>
>
> **The unified method for any detection head.** MC3D-Det offers multiple detection heads, including a combination of classification and regression (Centerpoint) and end-to-end (DETR) approaches. However, designing a uniform differentiable mechanism that satisfies 2D-3D consistency is challenging due to the diverse nature of these detection heads. For instance, in Centerpoint, the final prediction is obtained by combining the outputs of the classification head and an offset regression head. It is difficult to effectively supervise the classification head in this case. In contrast, our approach presents a unified solution through a rendering approach.
>
>
> **Significant improvement on both source and target domains.** Our approach is highly effective in improving the performance of both the source and target domains. In the source domain, our approach utilizes rendering to generate new views with different camera parameters, enabling the network to learn perspective-independent features.  In the target domain, our approach employs a more robust 2D detector to correct the 3D results by rendering. The other 2D-3D consistency methods do not have much effect in the source domain because 3D supervision is inherently more accurate and informative. However, our approach stands out by enabling the supervision of new perspectives through rendering.
>
>
> [1] Yang J, Wang T, Ge Z, et al. Towards 3D Object Detection with 2D Supervision. arXiv preprint arXiv:2211.08287, 2022.
> [2] Lian Q, Xu Y, Yao W, et al. Semi-supervised monocular 3d object detection by multi-view consistency. European Conference on Computer Vision. Cham: Springer Nature Switzerland, 2022: 715-731.
>
>
> We sincerely appreciate your constructive suggestions, and we will incorporate the corresponding experiments and additional explanations into the paper to improve its quality. We also look forward to further discussions with you. We also hope that you will consider revising the grading of our paper if you are satisfied with our response. Thank you again for your valuable input.

---

> ### Author Response · Authors · 2023-11-21
> **Further discussion**
>
> Dear reviewer kBUt,
>
> As the window for reviewer-author interaction is closing soon, I wanted to extend my sincerest gratitude for the invaluable time and effort you have dedicated to reviewing our work. To ensure that we have met your expectations, may I kindly ask if you find our responses satisfactory and if there are any remaining issues that need further clarification or improvement?

---

### Official Review · Reviewer_tA1k · 2023-10-31

**Soundness:** 3 good
**Presentation:** 3 good
**Contribution:** 2 fair
**Rating:** 5
**Confidence:** 5

**Summary:**

This paper introduces a method (MC3D-Det) to solve domain shift problem in multi-view 3D object detection. The proposed method aims to tackle this problem by aligning 3D detection with 2D detection results to ensure accurate detections. The framework, grounded in perspective debiasing, enables the learning of features that are resilient to changes in domain. It renders diverse view maps from bird's eye view features and corrects the perspective bias of these maps. Experimental results prove its efficiency in both Domain Generalization (DG) and Unsupervised Domain Adaptation (UDA).

**Strengths:**

*  Experiments illustrate that the proposed approach outperforms previous approaches （DG-BEV） on nuScenes dataset.
*  The paper is well written, and comprehensive component analysis.

**Weaknesses:**

* This article corrects the model's bias through the consistency of 2D detection and 3D detection. I am quite curious whether the 2D render is necessary. Is it possible to project the 3D box into a 2D plane and supervision only applied to the 2D bounding box. There are many papers 3D consistency supervision on monocular 3D detection. From this perspective, the novelty of the model is insufficient.

* How to evaluate the quality of the 2D branch of the render, at first I thought the model render the rgb image, but after carefully reading the paper, I found it was mainly about the heatmap. However, from Figure 3 (c), it does not show the quality of the rendered heatmap very well. So, how do we validate the motivation well.

* The proposed method has limitations as it has not been validated on sparse-query methods. In the past year, sparse-query methods like Sparse4Dv2, SparseBEV have shown great performance and speed advantages. Without an explicit BEV, would the domain shift problem still be as significant?

**Questions:**

*  The 3D consistency supervision of bounding boxes is need to validate the motivation.
*  More experiments about sparse-query methods are needed to prove the effectiveness of the method.

---

> ### Author Response · Authors · 2023-11-19
> **The first part of the response to reviewer tA1k**
>
> We sincerely appreciate the time and effort the reviewer dedicated to reviewing our paper and providing constructive comments. Below are our responses to the raised concerns.
>
>
> # **1. The necessity of the rendering**
>
> The key difference between our render-based method and existing 2D-3D consistency approaches lies in the following aspects:
>
>
> **Geometric feature correction with less semantic destruction.** Our rendering approach enables accurate projection of 2D images to the exact location of the BEV, thereby allowing the network to modify the geometric position of features with minimal semantic destruction. In contrast, 2D-3D consistency constraints have a significant impact on the entire network and often result in the destruction of semantic information. To compare our approach with existing 2D-3D consistency methods, we selected two representative approaches, namely Hybrid-Det [1] and MVC-Det[2]. Specifically, we report the unsupervised domain adaptation result (mAP/NDS) of these methods in nuScenes (source) and Lyft (target) datasets:
>
>
> | Method| nuScenes (source) | Lyft (target) |
> | :---:  | :---:  |:---:  |
> | DG-BEV |0.330  / 0.397 | 0.284 / 0.435 |
> | MVC-Det | 0.314 / 0.374 | 0.288 / 0.437 |
> | Hybrid-Det | 0.301 / 0.361 | 0.292 / 0.441 |
> | Ours | 0.331 / 0.396 | 0.316 / 0.476 |
>
>
> As shown in the table above, our algorithm achieves excellent performance. Although MVC-Det and Hybrid-Det exhibit improved performance in the target domain, their results in the source domain have deteriorated significantly. This is because other 2D-3D consistency methods enable them to fine-tune the entire network with semantic destruction. In contrast, our algorithm modifies the geometric space position of the feature, which leads to relatively stable results in the source domain.
>
>
> **The unified method for any detection head.** MC3D-Det offers multiple detection heads, including a combination of classification and regression (Centerpoint) and end-to-end (DETR) approaches. However, designing a uniform differentiable mechanism that satisfies 2D-3D consistency is challenging due to the diverse nature of these detection heads. For instance, in Centerpoint, the final prediction is obtained by combining the outputs of the classification head and an offset regression head. It is difficult to effectively supervise the classification head in this case. In contrast, our approach presents a unified solution through a rendering approach.
>
>
> **Significant improvement on both source and target domains.** Our approach is highly effective in improving the performance of both the source and target domains. In the source domain, our approach utilizes rendering to generate new views with different camera parameters, enabling the network to learn perspective-independent features.  In the target domain, our approach employs a more robust 2D detector to correct the 3D results by rendering. The other 2D-3D consistency methods do not have much effect in the source domain because 3D supervision is inherently more accurate and informative. However, our approach stands out by enabling the supervision of new perspectives through rendering.
>
>
> [1] Yang J, Wang T, Ge Z, et al. Towards 3D Object Detection with 2D Supervision. arXiv preprint arXiv:2211.08287, 2022.
>
> [2] Lian Q, Xu Y, Yao W, et al. Semi-supervised monocular 3d object detection by multi-view consistency. European Conference on Computer Vision. Cham: Springer Nature Switzerland, 2022: 715-731.
>
>
> # **2. The quality of the heatmap**
>
>
> Figure 3 shows that a more robust 2D detector can be used to correct spurious BEV geometry by correcting the rendered 2D heatmaps. In fact, Fig. 3 (d) is the heatmaps of our method, while Fig. 3 (c) is the heatmaps affected by domain shift without using our method. Specifically, Fig. 3 (c) shows that the rendering heatmaps of the target domain deteriorate due to domain shift, indicating that the chaotic BEV geometry. After correction by our algorithm, Figure 3 (d) shows more accurate rendering heatmaps on the target domain. In this way, our algorithm can use more robust 2D detection results to correct 3D chaotic geometric distributions. Our ultimate goal is to improve the detection performance, so we do not have quantitative statistical analysis on the heatmaps.

---

> ### Author Response · Authors · 2023-11-19
> **The second part of the response to reviewer tA1k**
>
> # **3. More experiments about sparse-query**
>
> We also conducted experiments on several methods without BEV representation (DETR3D [1], PETR [2], SparseBEV [3]) on cross-domain results (from nuScenes to Lyft), as shown in the table below.
>
> | Method | w/o ours | w ours |
> | :---:   |:---:  | :---:  |
> |         | **mAP/NDS**| **NDS/NDS**|
> | DETR3D$^*$      | 0.008/ 0.044 |0.028/ 0.076 |
> | PETR$^*$      | 0.012/ 0.051 |0.032/ 0.091 |
> | SparseBEV$^*$ | 0.016/ 0.059  |0.038/ 0.0097 |
> | BEVFormer| 0.084 / 0.246 | 0.208 / 0.355 |
> | BEVDet     | 0.104 / 0.275 | 0.296 / 0.446 |
> | FB-BEV    | 0.113/ 0.294 | 0.301 / 0.454|
> | BEVDepth  | 0.114 / 0.296 | 0.304/ 0.458 |
>
> Methods with an asterisk ($^*$) do not have BEV representation, while all others do. For these methods without BEV representation, we leverage the LSS mechanism in BEVDet [4] to build additional BEV presentation so that our algorithm can be used to improve these methods. Notably, our algorithm is model-agnostic, in other words, our algorithm can be applied to any method with image features and BEV features as shown in Fig. 2. According to the table, we draw a number of conclusions:
>
>
> (1) Methods without BEV presentation perform very poorly across domains compared to other algorithms. This can be attributed to their tendency to overfit camera extrinsic parameters in a learning way. Conversely, methods with BEV representation solely employ camera extrinsic parameters to project 2D image features into 3D space through a physical modeling form. This approach is highly resilient to variations in camera extrinsic parameters, thereby increasing its robustness and reliability.  In conclusion, the BEV representation can effectively establish the connection of different perspectives through physical modeling, as opposed to learning. This way enables the model to have superior cross-domain generalization.
>
>
> (2) Our algorithm has the potential to enhance the performance of DETR3D, PETR, and SparesBEV algorithms by rendering the new viewpoints from an auxiliary BEV presentation. This is because the process of rerendering new perspectives compels the network to acquire a generalizable BEV representation. The generalizable BEV representation, in turn, encourages the network to learn more robust visual features that mitigate the impact of overfitting camera parameters.
>
>
> [1] Wang Y, Guizilini V C, Zhang T, et al. Detr3d: 3d object detection from multi-view images via 3d-to-2d queries. Conference on Robot Learning. PMLR, 2022: 180-191.
>
> [2] Liu Y, Wang T, Zhang X, et al. Petr: Position embedding transformation for multi-view 3d object detection. European Conference on Computer Vision. Cham: Springer Nature Switzerland, 2022: 531-548.
>
> [3] Liu H, Teng Y, Lu T, et al. Sparsebev: High-performance sparse 3d object detection from multi-camera videos. Proceedings of the IEEE/CVF International Conference on Computer Vision. 2023: 18580-18590.
>
> [4] Huang J, Huang G, Zhu Z, et al. Bevdet: High-performance multi-camera 3d object detection in bird-eye-view. arXiv preprint arXiv:2112.11790, 2021.

---

> ### Author Response · Authors · 2023-11-19
> **The third part of the response to reviewer tA1k**
>
> # **4. The contribution of our paper**
>
>
> We wish to emphasize that our paper offers more than just a set of algorithms to improve the generalization performance of MC3D-Det. In fact, we are the first to explore unsupervised domain adaptation (UDA) on MC3D-Det and the first sim2real on the MC3D-Det task, which significantly contributes to this field.
>
> **The research significance of UDA on MC3D-Det.** In real-world scenarios, pre-trained models may only be available for certain scenarios and vehicle states, while unlabeled data may be available for other scenarios and vehicles. Improving the model's performance under this protocol is crucial, and we are the first to explore this problem. Additionally, we reproduce common UDA algorithms such as pseudo-label and feature alignment methods (Coral and AD), providing valuable insights and contributions to the field.
>
> **The research significance of sim2real on MC3D-Det.** Closed-loop testing are critical for autonomous driving.  Virtual engines (such as Carla) provide a feasible solution for closed-loop training, as they can synthesize a variety of extreme situations to improve the safety of driverless vehicles and reduce the annotation requirements [1, 2, 3]. Currently, there is only one paper that completes the closed-loop train, which must be in the virtual engine[4]. However, the domain gap between virtual engines and real scenes must be bridged. Our paper contributes to the field by exploring sim2real on the MC3D-Det task, providing valuable insights and paving the way for future research in this area.
>
> [1] Li H, Sima C, Dai J, et al. Delving into the Devils of Bird's-eye-view Perception: A Review, Evaluation and Recipe. IEEE Transactions on Pattern Analysis and Machine Intelligence, doi: 10.1109/TPAMI.2023.3333838.
>
> [2] Ma Y, Wang T, Bai X, et al. Vision-centric bev perception: A survey. arXiv preprint arXiv:2208.02797, 2022.
>
> [3] Wang T, Kim S, Ji W, et al. DeepAccident: A Motion and Accident Prediction Benchmark for V2X Autonomous Driving. arXiv preprint arXiv:2304.01168, 2023.
>
> [4] Jia X, Gao Y, Chen L, et al. Driveadapter: Breaking the coupling barrier of perception and planning in end-to-end autonomous driving[C]//Proceedings of the IEEE/CVF International Conference on Computer Vision. 2023: 7953-7963.
>
>
>
> We sincerely appreciate your constructive suggestions, and we will incorporate the corresponding experiments and additional explanations into the paper to improve its quality. We also look forward to further discussions with you. We also hope that you will consider revising the grading of our paper if you are satisfied with our response. Thank you again for your valuable input.

---

> ### Author Response · Authors · 2023-11-21
> **Further discussion**
>
> Dear reviewer tA1k,
>
> As the window for reviewer-author interaction is closing soon, I wanted to extend my sincerest gratitude for the invaluable time and effort you have dedicated to reviewing our work. To ensure that we have met your expectations, may I kindly ask if you find our responses satisfactory and if there are any remaining issues that need further clarification or improvement?

---

### Official Review · Reviewer_HYKF · 2023-11-08

**Soundness:** 3 good
**Presentation:** 4 excellent
**Contribution:** 3 good
**Rating:** 8
**Confidence:** 5

**Summary:**

The manuscript introduces a method designed to enhance feature learning in a way that is robust to changes in domain, leveraging an approach centered around perspective debiasing. It substantiates its efficacy through experimental findings in the contexts of Domain Generalization and Unsupervised Domain Adaptation.

**Strengths:**

Firstly, the proposed framework is innovative and can be smoothly incorporated into existing 3D detection techniques.
Secondly, the breadth of the experiments conducted is comprehensive, effectively illustrating the framework's robustness and effectiveness.

**Weaknesses:**

Figure 1 presents some ambiguity. It is not immediately clear how this figure is intended to convey the robustness of the perspective view to domain shifts.

**Questions:**

An inquiry for further clarification: How would methods that do not employ perspective view transformations, such as DETR3D, fare in performance across varying domains?

---

> ### Author Response · Authors · 2023-11-16
> **The response to reviewer HYKF**
>
> Thank you for your valuable feedback and acknowledgement.
>
>
> # **1. More explanation for Figure 1**
>
>
>
> Figure 1 does not depict our specific approach or demonstrate how to convey the robustness of the perspective view to domain shifts. Its purpose is merely to illustrate our motivation: we reveal a considerable discrepancy between the 3D prediction results and the ground-truth, which is the domain gap arising from overfitting with limited viewpoints, camera parameters, and similar environments. However, the target domain lacks additional supervision to rectify this error. Fortunately, we have identified that 2D detectors exhibit remarkable cross-domain performance. Consequently, we utilize more robust 2D results to constrain the new rendered perspective and rectify the spurious BEV features in the target domain. Furthermore, we propose to augment the network's capacity to learn more robust BEV features by rendering 2D images from different perspectives in the source domain. In summary, we establish the relationship between BEV features and 2D images through a neural rendering method, which significantly enhances the generalization ability of the detection results.
>
>
> # **2. More methods for cross-domain testing**
>
>
> We also conducted experiments on several methods without BEV representation (DETR3D [1], PETR [2], SparseBEV [3]) on cross-domain results (from nuScenes to Lyft) as shown in the table below.
>
> | Method | w/o ours | w ours |
> | :---:   |:---:  | :---:  |
> |         | **mAP/NDS**| **NDS/NDS**|
> | DETR3D$^*$      | 0.008/ 0.044 |0.028/ 0.076 |
> | PETR$^*$      | 0.012/ 0.051 |0.032/ 0.091 |
> | SparseBEV$^*$ | 0.016/ 0.059  |0.038/ 0.0097 |
> | BEVFormer| 0.084 / 0.246 | 0.208 / 0.355 |
> | BEVDet     | 0.104 / 0.275 | 0.296 / 0.446 |
> | FB-BEV    | 0.113/ 0.294 | 0.301 / 0.454|
> | BEVDepth  | 0.114 / 0.296 | 0.304/ 0.458 |
>
> Methods with an asterisk ($^*$) do not have BEV representation, while all others do. For these methods without BEV representation, we leverage the LSS mechanism in BEVDet [4] to build additional BEV presentation so that our algorithm can be used to improve these methods. Notably, our algorithm is model-agnostic, in other words, our algorithm can be applied to any method with image features and BEV features as shown in Fig. 2. According to the table, we draw a number of conclusions:
>
>
> (1) Methods without BEV presentation perform very poorly across domains compared to other algorithms. This can be attributed to their tendency to overfit camera extrinsic parameters in a learning way. Conversely, methods with BEV representation solely employ camera extrinsic parameters to project 2D image features into 3D space through a physical modeling form. This approach is highly resilient to variations in camera extrinsic parameters, thereby increasing its robustness and reliability.  In conclusion, the BEV representation can effectively establish the connection of different perspectives through physical modeling, as opposed to learning. This way enables the model to have superior cross-domain generalization.
>
>
> (2) Our algorithm has the potential to enhance the performance of DETR3D, PETR, and SparesBEV algorithms by rendering the new viewpoints from an auxiliary BEV presentation. This is because the process of rerendering new perspectives compels the network to acquire a generalizable BEV representation. The generalizable BEV representation, in turn, encourages the network to learn more robust visual features that mitigate the impact of overfitting camera parameters.
>
>
> [1] Wang Y, Guizilini V C, Zhang T, et al. Detr3d: 3d object detection from multi-view images via 3d-to-2d queries. Conference on Robot Learning. PMLR, 2022: 180-191.
>
> [2] Liu Y, Wang T, Zhang X, et al. Petr: Position embedding transformation for multi-view 3d object detection. European Conference on Computer Vision. Cham: Springer Nature Switzerland, 2022: 531-548.
>
> [3] Liu H, Teng Y, Lu T, et al. Sparsebev: High-performance sparse 3d object detection from multi-camera videos. Proceedings of the IEEE/CVF International Conference on Computer Vision. 2023: 18580-18590.
>
> [4] Huang J, Huang G, Zhu Z, et al. Bevdet: High-performance multi-camera 3d object detection in bird-eye-view. arXiv preprint arXiv:2112.11790, 2021.

---

> ### Author Response · Authors · 2023-11-21
> **Further discussion**
>
> Dear reviewer HYKF,
>
> As the window for reviewer-author interaction is closing soon, I wanted to extend my sincerest gratitude for the invaluable time and effort you have dedicated to reviewing our work. To ensure that we have met your expectations, may I kindly ask if you find our responses satisfactory and if there are any remaining issues that need further clarification or improvement?

---

### Official Review · Reviewer_seTP · 2023-11-09

**Soundness:** 3 good
**Presentation:** 1 poor
**Contribution:** 3 good
**Rating:** 6
**Confidence:** 3

**Summary:**

The paper considers the task of 3D object detection from multiple cameras. Existing methods exhibit poor generalization due to overfitting to specific viewpoints and environments. This paper proposes to re-render heatmaps between 2D views using an implicit 3D representation to learn features that are independent of the perspective and context. The paper demonstrates strong quantitative results on the task of domain generalization as well as the newly proposed task of unsupervised domain adaptation.

**Strengths:**

* The task considered is highly important. 3D object detection should generalize across different camera setups and environments
* Paper demonstrates strong quantitative results. Detections look qualitatively more accurate than existing baselines.

**Weaknesses:**

* The paper has clarity issues that make it difficult to understand. It is not clear what the perspective bias comes from in the first place, and how re-rendering different viewpoints addresses the issues of poor generalization. It is also not clear how the perspective debiasing works
* Figure 3, the only figure in the paper that explains the perspective debiasing, is uninformative. It’s not clear how the re-rendering has improved the heatmaps.

**Questions:**

* How are the features warped between images?
* Which components of the model need to be retrained, and which ones are lifted off-the-shelf from existing BEV methods? Are they finetuned?
* How does the network learn the depth and heights given new target domains without 3D data?

---

> ### Author Response · Authors · 2023-11-15
> **The first response part to reviewer seTP**
>
> I am very sorry for your confusion about my paper, which may be caused by your lack of knowledge about our field. This is my further elaboration of our paper, hoping to help you understand our paper.
>
> # **1. What is the source of perspective bias in MC3D-Det?**
>
> The perspective bias in MC3D-Det comes from overfitting with limited viewpoints, camera parameters, and similar environments. The perspective bias represents the error of the final 3D result projected in a single viewing (2D image plane).
>
>
> # **2. How does re-rendering address the issues of poor generalization?**
>
>
> We restate our method from three aspects: the specific approach, the in-depth reason, and the mathematical derivation.
>
>
> **The specific approach**: we establish an implicit foreground volume (IFV) to associate the camera with the bird's eye view (BEV) plane and re-render different view maps from BEV features. IFV allows for rendering view maps from BEV features with different camera parameters. By supervising the rendering of maps from different viewing angles, BEV features can be more accurately placed in the exact location. For the target domain, we can use 2D detectors trained in the source domain to monitor the rendered view so that the BEV features fall to the exact location.
>
>
> **The in-depth reason**: (1) For the source domain, supervising new view maps with different camera parameters can help the network learn perspective-invariant features. Because only the features of the 2D image are placed in the exact 3D position, we can re-render the reasonable result in a new viewpoint. (2) For the target domain, the more robust 2D detector can correct 3D results by reprojecting on a single image. This is because domain offset allows the network to place the 2D image in the wrong 3D position, which can be corrected by rendering consistency.
>
>
> **The mathematical derivation**: The 3D prediction error caused by domain shift can be reflected in a single perspective, and the final error on the single perspective is defined as Eq. (2). Detailed proof can be found in Appendix C. In other words, correcting errors in different 2D planes can reduce the errors in the final 3D prediction results, which provides a theoretical basis for our method.
>
>
> # **3. How does perspective debiasing work?**
>
> We expound the principle of perspective debiasing on both the source domain and target domain:
>
> For the source domain, supervising new view maps with different camera parameters can help the network learn perspective-invariant features. Because only the features of the 2D image are placed in the exact 3D position, we can re-render the reasonable result in a new viewpoint. By generating diverse view maps from bird's eye view (BEV) features using implicit foreground volumes (IFV) to relate the camera and BEV planes, the MC3D-Det framework can render view maps with varied camera parameters, which helps the network learn perspective- and environment-independent features. This enhances the model's robustness to different perspectives and domain conditions, improving its generalization ability on the source domain.
>
> For the target domain, the more robust 2D detector can correct 3D results by reprojecting on a single image. This is because domain offset allows the network to place the 2D image in the wrong 3D position, which can be corrected by rendering consistency. By projecting the object's box from the ego coordinate to the 2D camera plane, generating category heatmaps and object dimensions, and using focal loss and L1 loss to supervise the class information and object dimensions on the source domain, the MC3D-Det framework can rectify the perspective bias of the view maps and learn the object heights and dimensions without requiring 3D data on the target domain. Additionally, a 2D detector is trained for the image feature using 3D boxes, and the depth of the network prediction is forced to learn normalized virtual depth to improve performance. This enables the network to learn the depth information without requiring explicit 3D data on the target domain.
>
> Overall, perspective debiasing addresses the issues of limited viewpoints, camera parameters, and similar environments, enhancing the model's generalization ability on both the source and target domains. By re-rendering diverse view maps and rectifying perspective bias, the model can learn perspective- and environment-independent features, improving its ability to generalize to new and unseen scenarios.

---

> ### Author Response · Authors · 2023-11-15
> **The second response part to reviewer seTP**
>
> # **4. How does the re-rendering improve the heatmaps?**
>
> Re-rendering is actually about fixing the messy geometry of the BEV space, in other words, putting 2D features in the right place. When 2D object features are placed in the wrong position, the rendered 2D heatmaps will be misrepresented. By monitoring the rendered heatmaps, it is to correct the incorrect geometric distribution in the BEV feature space.
>
> We visualized all kinds of heatmaps on the target domain as shown in Fig.3: (a) ground-truth, (b) 2D detector, (c) rendered from IVF, and (d) revised by 2D detector.  The heatmaps in Fig.3 (a) show the real annotation projected onto the 2D plane. The heatmaps in Fig.3 (b) are from 2D detectors trained in the source domain. The heatmaps in Fig.3 (c) are derived from the rendering of BEV features without $L_{con}$. The heatmaps in Fig.3 (D) are derived from the rendering of BEV features with $L_{con}$.
>
>
>
> # **5. How are the features warped between images?**
>
> Most algorithms lift 2D image features into 3D BEV space based on camera parameters, For details, see BEVDet [1], BEVFormer [2], and FB-BEV [3]. Notably, our algorithm is model-agnostic, which can be applied to various MC3D-Det algorithms. As shown in Figure 2, our method can be regarded as an additional constraint on the image feature and BEV feature, in other words, our algorithm can be applied to any method with image features and BEV features, such as BEVDet, BEVFormer, and FB-BEV, as provided in Table 3. Although many additional auxiliary networks are used in this process, none of these networks participate in the reasoning process.
>
> [1] Huang J, Huang G, Zhu Z, et al. Bevdet: High-performance multi-camera 3d object detection in bird-eye-view. arXiv preprint arXiv:2112.11790, 2021.
>
> [2] Li Z, Wang W, Li H, et al. Bevformer: Learning bird’s-eye-view representation from multi-camera images via spatiotemporal transformers. European conference on computer vision. Cham: Springer Nature Switzerland, 2022: 1-18.
>
> [3] Li Z, Yu Z, Wang W, et al. Fb-bev: Bev representation from forward-backward view transformations. Proceedings of the IEEE/CVF International Conference on Computer Vision. 2023: 6919-6928.
>
> # **6. Training details**
>
> We retrain these detectors following their default settings and add the proposed constraints as training-only auxiliary networks and losses. Therefore, all components are retrained, but it is not a finetuning process. As shown in Figure 2, our algorithm only imposes additional constraints on image features and BEV features through some auxiliary networks. So, compared to the original algorithm, we just add additional loss and auxiliary networks during training. In Figure 2, the main pipeline is the existing MC3D-Det algorithm, and the perspective debiasing above is our proposed regular constraint. Not a fine-tuning process, but a complete training process with the auxiliary network and additional loss.
>
>
> # **7. How do we learn the depth and heights without 3D data on target domains?**
>
> Due to domain migration, BEV features are often confused in the target domain, and the rendered 2D heatmaps are inaccurate. We use more robust 2D detectors to correct these BEV features at different viewing angles. Through this mechanism, BEV features can be better modified in the target domain. As shown in Fig.4 (b), on the target domain, the 2D detector has good generalization performance and can accurately detect the center of the object. However, the heatmap rendered from the IFV, without refinement, is very spurious and it is very difficult to find the center of the objectin in Fig.4 (c). Fig.4 (d) shows that rendered heatmaps of IFV can be corrected effectively with 2D detectors.

---

> ### Author Response · Authors · 2023-11-21
> **Further discussion**
>
> Dear reviewer seTP,
>
> As the window for reviewer-author interaction is closing soon, I wanted to extend my sincerest gratitude for the invaluable time and effort you have dedicated to reviewing our work. To ensure that we have met your expectations, may I kindly ask if you find our responses satisfactory and if there are any remaining issues that need further clarification or improvement?

---

> ### Comment · Reviewer_seTP · 2023-11-21
>
> I thank the authors for the detailed clarifications and additional experiments. In light of these clarifications, I have bumped up my rating.

---

> > ### Author Response · Authors · 2023-11-22
> >
> > Dear reviewer seTP,
> >
> > We express our sincere appreciation for your positive affirmation, which is a significant source of encouragement. Additionally, we extend our gratitude for your diligent efforts and hard work.

---

### Official Review · Reviewer_S4As · 2023-11-09

**Soundness:** 3 good
**Presentation:** 2 fair
**Contribution:** 3 good
**Rating:** 6
**Confidence:** 4

**Summary:**

- The paper focuses on the problem of multi-view 3D object detection for autonomous driving conditions.
- The main focus is when there is a domain gap between training (source) and testing (target_ conditions.
- The authors evaluate two scenarios: a) domain generalization, i.e., no target images available, and b) unsupervised domain adaptation, i.e., target images available (no labels, of course).
- Key Idea: Sec. 3.2 shows that the existing methods overfit to camera intrinsics and extrinsics of the train images - perspective bias of the model. The proposed method MC3D-Det modifies the BEVDepth pipeline using perspective debiasing to fix this.
- What is perspective debiasing? In the source domain, perturb the existing camera views and render the "view maps" from the novel camera views. These random camera positions and angles help to avoid overfitting.
- Decreasing the domain gap? If you have unlabelled target images, MC3D-Det uses an offshelf single view 2D detector to predict 3D bboxes and use it to rectify the BEV features using consistency loss.
- Evaluation Datasets: nuScenes (real), Lyft (real), DeepAccident (synthetic). Metrics are standard 3D bbox detection metrics.
- Baselines: BEVDepth (the base method used by MC3D-Det), DG-BEV. Other domain adapation baselines like Pseudo labeling, Oracle etc.
- Table 1. shows that the proposed method improved mAP on the target domain by about 2-4% over the baseline.

**Strengths:**

- The main idea is presented clearly, and the proposed approach is intuitive and easy to understand. The technical details are all laid out in Sec. 4 and the supplementary. The technical contributions made by MC3D-Det are novel.
- The perspective bias of the model, Eq. 2, is derived theoretically in Sec. 3 and supplementary using limited assumptions and first principles. This stands at the core of the motivation of the proposed approach and is, therefore, an important step.
- Substantial evaluations are done on multiple datasets (Lyft, nuScenes, DeepAccident), especially cross-domain evaluations, including pseudo labeling and oracle training. The method is compared against multiple relevant baselines.
- The ablative studies in Table 2 and Table 3 are informative and highlight the importance of source and target domain debiasing and the plug-and-play capabilities of the MC3D-Det with existing methods.
- Main results show gains of the proposed method for domain generalization and unsupervised domain adaptation.

**Weaknesses:**

- Missing details on BEVDepth architecture used on Table. 1: The performance of the baseline BEVDepth reported on the source domain nuScenes (I am assuming this is val set, the argument also holds for the test set) in Table. 1 of 32.6 mAP is significantly less than the published results of BEVDepth (refer Table. 7) of 41.8 mAP, R101-DCN architecture. Bigger backbones with the scale of data like nuScenes exhibit better generalization; it is worth while investigating if the performance gain of the proposed MC3D-Det also holds when using BEVDepth at its full capacity. Especially, since BEVDepth is the base method for MC3D-Det.

Quick minor comment on similar lines on DG-BEV. The reproduced results for DG-BEV in Table. 1 are much worse than the published results of DG-BEV (Table. 1). For Lyft -> nuScenes, for domain generalization, DG-BEV (published) == MC3D-Det > DG-BEV (reproduced). However, the code for DG-BEV is not available, so I would side with the authors here.

- Computational overhead compared to baselines: Proposed method adds an overhead to the baseline method. It would be helpful to quantify the additional parameters and GLOPs used compared to the methods evaluated in Table. 3.

- The focus on the car category: Please mention the categories used for evaluation in Table. 1. Correct me if I am wrong, but all the evaluations only consider the vehicle and car category (Lyft <-> nuScenes) - a rigid object with easy-to-learn 3D geometry prior, allowing for a consistent rendering with a perturbed camera. Do the results also hold for categories like the pedestrian, cyclists or less represented rigid classes like truck, construction vehicle, bus, and trailer? Using Waymo -> nuScenes protocol here would be more informative.

- Camera perturbations used: The qualitative results do not show the rendered views from the perturbed cameras. Since perspective overfitting is an issue, we should augment the view to avoid biasing, but still, we want to stay within the camera extrinsics distributions of the target domain. How is this balance achieved? What is the magnitude of translation and rotation perturbations used for perspective debiasing in MC3D-Det? How does the performance change when the perturbations increase from the anchor positions?

Minor:
- Sec. 1. However, without taining data -> However, without training data
- Please increase text font in Figure 2. Mention the intermediate feature in the figure using notation established.
- Sec 4.1, C, X, Y, Z are not defined.
- Eq 6: D_vitural -> D_virtual
- Sec. 4.3. there is not 3D labeled -> there are no 3D labels
- Sec 5.1. It demonstrate that -> It demonstrates that
- Sec 5.3. 2D Detetctor -> 2D Detector
- Table 2, caption: 2D Detetctor -> 2D Detector
- Fig. 3: The bounding boxes and the image is extremely small. The image should be visible without zooming in.
- Sec. 6: or 2D pre-trained 2D detectors -> or pre-trained 2D detectors

**Questions:**

As above.
- Fair baseline comparison to BEVDepth.
- Parameter overhead.
- Generalization to other categories.
- Information on Camera perturbations.

---

> ### Author Response · Authors · 2023-11-14
> **The first response part to reviewer S4As**
>
> I would like to express my gratitude for your invaluable comments, particularly for pointing out numerous typos, which will significantly enhance the quality of my paper.
>
> # **1. The stronger baseline**
>
> We improve our baseline by changing the backbone to R101-DCN and the input resolution to 768 $\times$ 1408. Additionally, both our proposed method and DG-BEV modify the basic backbone and image resolution. These approaches are tested on a cross-dataset (from nuScenes to Lyft), and the results for the source domain (nuScenes) and target domain (Lyft) are presented in the table below:
>
>
> | Method |Protocol| nuScenes (mAP / NDS) | Lyft (mAP / NDS) |
> | :---:  | :---:  |:---:  | :---:  |
> | BEVDepth| DG  |0.414 / 0.538 | 0.120 / 0.304 |
> | DG-BEV | DG  |0.409 / 0.531 | 0.279 / 0.431 |
> | Ours | DG  | 0.421 / 0.541 | 0.324 / 0.484 |
> | Ours |  UDA  | 0.417 / 0.539 | 0.335 / 0.497 |
>
> Here, DG represents domain generalization, and UDA represents unsupervised domain adaptation. As shown in the table, the results in the source domain (nuScenes) have significantly improved, but the improvement in the target domain (Lyft) is less pronounced. This indicates that increasing the model size and image resolution has limited effect on the model's cross-domain performance. It is worth noting that our proposed method still significantly improves the results in the target domain.
>
>
> # **2. Parameter overhead**
> Our algorithm does not add any inference overhead when reasoning. As shown in Figure 2, our algorithm adds auxiliary constraints on image features and BEV features during training. Although many additional auxiliary networks are used in this process, none of these networks participate in the reasoning process. In other words, the original structure of the network remains unchanged during inference. Moreover, our algorithm is model-agnostic, which can be applied to various MC3D-Det algorithms, such as BEVFormer and FB-BEV, as provided in Table 3.

---

> ### Author Response · Authors · 2023-11-14
> **The second response part to reviewer S4As**
>
> # **3. The results of other categories**
>
> We conduct domain generalization tests (from nuScenes to lyft) on more categories, and the results are presented in the following table:
>
> | Method | car | truck | bus | bicycle | pedestrian |
> | --- | --- | --- | --- | --- | --- |
> | BEVDepth | 0.103 | 0.000 | 0.000 | 0.000 | 0.004 |
> | DG-BEV | 0.263 | 0.000 | 0.012 | 0.027 | 0.021 |
> | Ours | 0.297 | 0.016 | 0.052 | 0.047 | 0.077 |
>
> The table reveals that the other categories exhibited a substandard performance across all methods. Notably, our algorithm exhibited enhanced performance in the aforementioned categories. This improvement can be attributed to the different viewpoint heatmaps for diverse categories, as opposed to the use of original images. Such an approach has demonstrated the algorithm's resilience to non-rigid bodies. The lackluster performance of other categories can be attributed to the substantial variance in image representation across distinct datasets.
>
>
>
> # **4. Information on Camera perturbations**
>
> We provide a systematic discussion on the perturbations of camera parameters, which play a vital role in semantic rendering. The extrinsic parameters include perturbations in rotation (pitch, yaw, roll) and translation, while the intrinsic parameters mainly concern the focal length.
>
> **The camera extrinsic parameters**
>
> The perturbed range of pitch, yaw, roll, are discussed in the supplementary appendix Fig.9. As shown in Fig.9, a small increase in the pitch and roll range is optimal, as most of the cameras on the car are parallel to the road with only slight fluctuations. However, yaw perturbations require a larger augmentation angle to facilitate the network in learning perspective-invariant features.
>
> Furthermore, we evaluate the perturbed range of translation (X, Y, Z), and the results are presented in the table above.
>
> | Perturbed X range (m)  | mAP | NDS |
> | --- | --- | --- |
> | [0.0, 0.0] | 0.290 | 0.438 |
> | [-1.0, 1.0] | 0.295 | 0.446 |
> | [-2.0, 2.0] | 0.301 | 0.453 |
> | [-4.0, 4.0] | 0.294 | 0.443 |
>
>
> | Perturbed Y range (m)  | mAP | NDS |
> | --- | --- | --- |
> | [0.0, 0.0] | 0.290 | 0.438 |
> | [-1.0, 1.0] | 0.296 | 0.444 |
> | [-2.0, 2.0] | 0.304 | 0.457 |
> | [-4.0, 4.0] | 0.300 | 0.449 |
>
>
> | Perturbed Z range (m)  | mAP | NDS |
> | --- | --- | --- |
> | [0.0, 0.0] | 0.290 | 0.438 |
> | [-1.0, 1.0] | 0.292 | 0.441 |
> | [-2.0, 2.0] | 0.294 | 0.443 |
> | [-4.0, 4.0] | 0.284 | 0.422 |
>
>
> The ablation in the table reveals that random translation of the observation position to render a new perspective can effectively enhance the model's performance. Notably, the difference between the camera extrinsic parameters of different datasets primarily pertains to the camera position and camera yaw angle. Among them, the camera position relative to the ego coordinate of car does not exceed 2m. Based on the findings presented in Figure 9 and the table above, it can be inferred that our proposed algorithm can significantly enhance the robustness of the model. In other words, our perturbation range already includes the camera extrinsic range of the target domain.
>
>
> **The camera intrinsic parameters**
>
> The camera intrinsic parameters (i.e., focal length) are affected by the resize amplitude of the input image, so we conducted an ablation experiment on this:
>
>
> | Perturbed resize range | mAP | NDS |
> | --- | --- | --- |
> | [-0.03, 0.06] | 0.274 | 0.399 |
> | [-0.06, 0.11] | 0.304 | 0.458 |
> | [-0.08, 0.18] | 0.299 | 0.447 |
> | [-0.14, 0.30] | 0.267 | 0.381 |
>
>
> Our experimental results reveal that significant perturbations in these camera intrinsic parameters can cause objects located at the edge of the image to be missed, ultimately leading to poor results. Therefore, we adopt the default resize range ([-0.06, 0.11]) to mitigate the aforementioned issue. Obviously, such a small resize range don't meet the alignment of intrinsic camera parameters (focal length 553 on nuScence and 407 on Lyft). In short, the camera intrinsic parameters cannot be aligned, but can also effectively improve the performance of the target domain.

---

> ### Author Response · Authors · 2023-11-21
> **Further discussion**
>
> Dear reviewer S4As,
>
> As the window for reviewer-author interaction is closing soon, I wanted to extend my sincerest gratitude for the invaluable time and effort you have dedicated to reviewing our work. To ensure that we have met your expectations, may I kindly ask if you find our responses satisfactory and if there are any remaining issues that need further clarification or improvement?

---

> > ### Comment · Reviewer_S4As · 2023-11-22
> >
> > Thank you for your work. My concerns are addressed - I would like to stick to my original rating.

---

### Meta-Review · Area_Chair_tLSn · 2023-12-09

**Metareview:**

This paper tackles the task of multi-camera 3D detection (in autonomous driving context), and in particular focuses on the task of transfer of trained systems to a new domain (with or without adaptation) and seeks to counter the effects of different intrinsics and extrinsics compared to the source domain. The are two key ideas that the approach builds on: a) ‘rendering’ the BEV features to obtain hetmaps from novel viewpoints (and supervising these in training) can enable the BEV features to be more robust, helping unsupervised domain transfer where the camera parameters may vary, b) given some unlabeled target domain data, this rendering can also help finetune the network by enforcing consistency with 2D detections from the raw images.

This paper received 5 reviews, with 1 reviewer recommending an accept (although with an unfortunately brief review) and others more borderline (3 leaning accept, 1 reject). The reviewers all appreciated the empirical results and the importance of the task tackled. However, there were some concerns raised (e.g. regarding clarity — and while the author responses did address these, the text is still slightly difficult to parse) and whether the novel-view rendering is critical (which the authors addressed via ablations and comparisons to baselines which do not perform such rendering).

Overall, this is a paper that tackles a relevant task and shows consistent improvements over prior work. However, while the improvements over a naive baseline are significant, the ones over ones tackling domain generalization are less so (e.g. 2-3% mAP over PD-BEV). Similarly, the idea of domain adaptation via a consistency loss only seems to yield minimal improvements (e.g. 1-2% mAP). Finally, the ablation reported in Tables 5-7 and Fig 9 also indicates that even not perturbing cameras can yield good performance (only 1-2% worse than the approach). These make it unclear how critical the two key ideas of i) camera perturbation, ii) domain adpation are!

These empirical gains, combined with an approach that is slightly complicated and tailored to a narrow task, as well as a presentation that is difficult to follow make the AC feel that this paper is perhaps just below the bar for ICLR and the decision is therefore to lean towards rejection.

**Justification For Why Not Higher Score:**

While the approach yields more robust detection compared to baselines, these gains are arguably small. Moreover, the ablations and results also raise questions about how important the two key ideas in the approach are. This, combined with an approach that is slightly complicated and tailored to a narrow task, as well as a presentation that is difficult to follow make the AC feel that this paper is perhaps below the bar for ICLR

**Justification For Why Not Lower Score:**

N/A

---

### Decision · Program_Chairs · 2024-01-16

Reject